# Homeostasis of protein and mRNA concentrations in growing cells

Jie Lin[1] & Ariel Amir [1]

Many experiments show that the numbers of mRNA and protein are proportional to the cell volume in growing cells. However, models of stochastic gene expression often assume constant transcription rate per gene and constant translation rate per mRNA, which are incompatible with these experiments. Here, we construct a minimal gene expression model to fill this gap. Assuming ribosomes and RNA polymerases are limiting in gene expression, we show that the numbers of proteins and mRNAs both grow exponentially during the cell cycle and that the concentrations of all mRNAs and proteins achieve cellular homeostasis; the competition between genes for the RNA polymerases makes the transcription rate independent of the genome number. Furthermore, by extending the model to situations in which DNA (mRNA) can be saturated by RNA polymerases (ribosomes) and becomes limiting, we predict a transition from exponential to linear growth of cell volume as the protein-to-DNA ratio increases.

---

[1] John A. Paulson School of Engineering and Applied Sciences, Harvard University, Cambridge, MA 02138, USA. Correspondence and requests for materials should be addressed to A.A. (email: arielamir@seas.harvard.edu)

Despite the noisy nature of gene expression[1–6], various aspects of single cell dynamics, such as volume growth, are effectively deterministic. Recent single-cell measurements show that the growth of cell volume is often exponential. These include bacteria[7–10], archaea[11], budding yeast[10,12–15] and mammalian cells[10,16]. Moreover, the mRNA and protein numbers are often proportional to the cell volume throughout the cell cycle: the homeostasis of mRNA concentration and protein concentration is maintained in an exponentially growing cell volume with variable genome copy number[17–22]. The exponential growths of mRNA and protein number indicate dynamical transcription and translation rates proportional to the cell volume, rather than the genome copy number. However, current gene expression models often assume constant transcription rate per gene and constant translation rate per mRNA (constant rate model)[1,5,23–25]. Assuming a finite degradation rate of mRNAs and non-degradable proteins, these models lead to a constant mRNA number proportional to the gene copy number and linear growth of protein number[26–28], incompatible with the proportionality of mRNA and protein number to the exponentially growing cell volume.

Since the cell volume, protein copy number and mRNA copy number grow exponentially throughout the cell cycle, one may expect a sufficient condition to achieve a constant concentration is to let them grow with the same exponential growth rate. However, mathematical analysis suggests this is insufficient. Let us consider the logarithm of protein concentration $c$, which can be written as $\ln(c) = \ln(p) - \ln(V)$. Here $p$ is the protein number and $V$ is the cell volume. If one assumes the protein number and the cell volume grow exponentially but independently, with time-dependent exponential growth rates $\lambda_p(t)$ and $\lambda_v(t)$ respectively, the time derivative of the logarithm of concentration then obeys $d\ln(c)/dt \sim \lambda_p(t) - \lambda_v(t)$. Even when the time-averaged growth rates of protein number and cell volume are equal, $\langle\lambda_p(t)\rangle = \langle\lambda_v(t)\rangle$, any fluctuations in the difference between them will accumulate and lead to a random walk behavior of the logarithm of concentration. The homeostasis of protein and mRNA concentrations implies that there must be a regulatory mechanism in place to prevent the accumulation of noise over time.

The main goal of this work is to identify such a mechanism by developing a coarse-grained model taking into account cell volume growth explicitly. Specially, we only consider continuously proliferating cells and do not take account of non-growing cells, e.g., bacterial cells in stationary phase[29]. The ubiquity of homeostasis suggests that the global machinery of gene expression, RNA polymerases (RNAPs) and ribosomes, should play a central role within the model. Based on the assumption that the number of ribosomes is the limiting factor in translation, we find that the exponential growth of cell volume and protein number originates from the auto-catalytic nature of ribosomes[30–33]. The fact that ribosomes make all proteins ensures that the protein concentrations do not diverge. Based on the assumption that the number of RNAP is the limiting factor in transcription, we find that the mRNA number also grows exponentially and the mRNA concentration is independent of the genome copy number because of the competition between genes for this global resource[18–20]. We also study the effects of genome replication. Due to the heterogeneous timing of gene replication, the transcription rate of one gene has a cell cycle dependence. Within our model, it doubles immediately after the gene is replicated and decreases gradually as other genes are replicated. Nevertheless, we find that this leads to a small effect on protein levels. Finally, we extend our model to more general situations in which an excess of RNAP (ribosome) leads to the saturation of DNA (mRNA). We propose a phase diagram of gene expression and cellular growth controlled by the protein-to-DNA ratio. We predict a transition from exponential growth to linear growth of cell volume as the protein-to-DNA ratio passes a threshold.

## Results

**Model of stochastic gene expression.** In constant rate models, the transcription rate per gene and the translation rate per mRNA are constant[1,5,24] (Fig. 1a). Constant rate models predict a constant mRNA number proportional to the gene copy number and independent of the cell volume. However, experimental observations on plant and mammalian cells have revealed a proportionality between mRNA number and cell volume for cells with a constant genome copy number[18–20]. Moreover, even

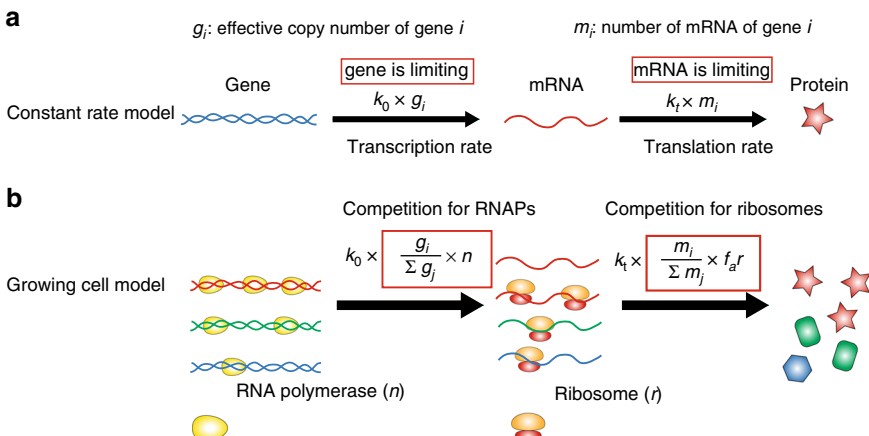

**Fig. 1** The growing cell model of stochastic gene expression in comparison with constant rate models. **a** In the constant rate model, the transcription rate is proportional to the gene copy number, and the translation rate is proportional to the mRNA number. These assumptions imply that the gene number and mRNA number are the limiting factors in gene expression. **b** In Phase 1 of the growing cell model, we introduce as limiting factors RNA polymerases (RNAPs) and ribosomes. Genes with different colors are transcribed with different rates. Here $k_0$ is a constant and the gene regulation is coarse-grained into the gene allocation fraction $\phi_i = g_i / \sum_j g_j$. $g_i$ is the effective copy number of gene $i$ (also accounting for the promoter strength). $n$ is the total number of RNAPs. Translation rates of mRNA depend on the number of active ribosomes ($f_a r$), the translation rate $k_t$, and the fraction of mRNA $i$ in the total pool of mRNA. In a later section (A unified phase diagram of gene expression and cellular growth), we will relax our assumptions and consider situations in which the limiting factors of gene expression become the gene number and the mRNA number

comparing the cells before and after the genome replication (S phase), the proportionality coefficient between mRNA and cell volume does not exhibit any obvious change. In contrast, a constant transcription rate per gene would predict a doubled transcription rate after the replication of the whole genome, leading to a higher mRNA concentration. In one class of constant rate models[26,27,34], a deterministic exponential growth of cell volume is explicitly considered. The resulting perturbation on the concentrations due to genome replication is suppressed in the long lifetime limit, but still significant for short lifetime molecules, e.g., mRNA (see Fig. 1 in ref.[27]).

Considering translation, various experiments have shown that the number of ribosomes is the limiting factor rather than the number of mRNAs. The most direct evidence is the growth law: the growth rate of cells is proportional to the fraction of ribosomal proteins in the total proteome (with a constant factor depending on the growth condition)[35] both for bacterial cells[30,31,36] and budding yeast cells[32]. This means a constant fraction of ribosomes are actively translating mRNAs. These results suggest that in general cells are below the saturation limit in which there are too many ribosomes that the mRNAs can bind. We will therefore assume the biological situation in which mRNAs in the cell compete for the limiting resource of actively translating ribosomes, therefore the translation rate of one type of mRNA is proportional to the number of active ribosomes times its fraction in the total pool of mRNAs.

Considering transcription, experiments have shown that mutants of fission yeasts altered in cell size regulated global transcription to maintain similar transcription rates per cell volume regardless of the cellular DNA content. The changes in total transcription correlated with coordinated changes in gene occupancy by RNA polymerases[37]. These results suggest that the number of RNAPs may be the limiting factor in transcription rather than the gene number, and similar evidence has been shown for bacterial cells[38] and mammalian cells[39]. However, in the same experiments on fission yeast[37], it has also been found that in cell-cycle-arrested mutants, total transcription rates stopped increasing as the cell volume exceeded a certain value, which suggested DNA became limiting for transcription at low DNA concentration. This result suggests that an excess of RNAPs may lead the gene number to become the limiting factor in certain conditions. In this section, we will focus on the scenario that both RNAP and ribosome are limiting in gene expression, which we denote as Phase 1. In this phase, we will show that the mRNA number and the protein number are proportional to the cell volume and grow exponentially. In a later section (A unified phase diagram of gene expression and cellular growth), we will consider a more general model in which the limiting nature of RNAPs and ribosomes may break down and the dynamics of mRNA and protein number is different.

To address the limiting nature of RNAP, we define an effective gene copy number $g_i$ for each gene to account for its copy number and the binding strength of its promoter, which determines its ability to compete for RNAPs. The transcription rate for one specific gene $i$ is proportional to the fraction of RNAPs that are working on its gene(s), $\phi_i = g_i / \sum_j g_j$, which we denote as the gene allocation fraction. Gene regulation is thus coarse-grained into the gene allocation fraction $\phi_i$. The transcription rate is independent of the genome copy number since a change in the genome number leaves the allocation fraction of one gene invariant, a conclusion which is consistent with a number of experimental results on various organisms[18–20,37].

In fact, explicit gene regulation can also be included in our model (Methods), with a time-dependent $g_i$. In such scenarios, $g_i$ may be a function of protein concentrations (for instance, the action of transcription factors modifies the transcription rate).

Such models will lead to more complex dynamics of mRNA and protein concentrations. However, since we are interested in the global behavior of gene expression and cell volume growth, we do not focus on these complex regulations in this manuscript. Our conclusions regarding the exponential growth of mRNA and protein number for constitutively expressed genes and the exponential growth of cell volume on the global level are not affected by the dynamics of gene expression of particular genes.

In the following, $m$, $p$, $r$, $n$ represents the numbers of mRNAs, proteins, ribosomes and RNA polymerases, respectively. Proteins ($p$) also include RNAPs ($n$) and ribosomes ($r$)[30]. We consider the degradation of mRNA with degradation time $\tau$ for all genes. The protein number decreases only through cell divisions (though adding a finite degradation rate for proteins does not affect our results). The stochastic dynamics of gene expression within Phase 1 of our model are summarized in the following sets of equations and Fig. 1b,

$$m_i \xrightarrow{k_0 \left( g_i / \sum_j g_j \right) n} m_i + 1, \tag{1}$$

$$m_i \xrightarrow{m_i/\tau} m_i - 1, \tag{2}$$

$$p_i \xrightarrow{k_t \left( m_i / \sum_j m_j \right) f_a r} p_i + 1. \tag{3}$$

Here $k_0$, $k_t$ are constants, characterizing the transcription (translation) rate of a single RNAP (ribosome). $f_a$ is the fraction of active ribosomes, which we assume to be constant in a given nutrient environment[30,32]. We note that nonspecifically bound RNAPs have been reported in bacteria[40,41]. We will discuss their effect later. For simplicity, we first assume the values of $\phi_i$ do not change in time. This can be formally thought of as corresponding to an instantaneous replication of the genome. In reality, a finite duration of DNA replication and the varying time of replication initiation for different genes lead to $\phi_i$'s that change during the DNA replication. We later analyze a more complete version of the model which includes these gene dosage effects, but we first consider the simplified scenario of constant $\phi_i$ that will capture the essential features of the problem.

We assume the cell volume is approximately proportional to the total protein mass, i.e., $V \propto M = \sum_j p_j$, which is a good approximation for bacteria[42,43] and mammalian cells[17]. To simplify the following formulas, we consider each protein has the same mass and set the cell density as 1.

Due to the fast degradation of mRNA compared with the cell cycle duration[44,45], the mRNA number can be well approximated as being in steady state. We can express the ensemble-averaged number of mRNA of gene $i$ as

$$\langle m_i(t) \rangle = k_0 \phi_i \langle n(t) \rangle \tau. \tag{4}$$

Equation (3) then leads to the time-dependence of average ribosome number, $d\langle r \rangle / dt = k_t f_a \phi_r \langle r \rangle$, reproducing the auto-catalytic nature of ribosome production and the growth rate

$$\mu = k_t f_a \phi_r, \tag{5}$$

determined by the relative abundance of active ribosomes in the proteome[30,32].

Similarly, the number of protein $i$ grows as $d\langle p_i \rangle / dt = k_t f_a \phi_i \langle r \rangle$. As the cell grows and divides, the dynamics becomes insensitive to the initial conditions, so the protein number will grow exponentially as well[21]. The ratio between the

averages of two protein numbers in the steady state is set by the ratio of their production rate, therefore $\langle p_i \rangle / \langle p_j \rangle = \phi_i / \phi_j$. The average number of mRNA traces the number of RNA polymerases according to Eq. (4), and therefore also grows exponentially. Throughout the cell cycle we have

$$\langle m_i(t) \rangle = m_b(i)\exp(\mu t), \tag{6a}$$

$$\langle p_i(t) \rangle = p_b(i)\exp(\mu t), \tag{6b}$$

where $m_b(i)$ ($p_b(i)$) is the number of mRNA (protein) of gene $i$ at cell birth.

We denote the concentrations of mRNA and protein of gene $i$ as $c_i^m = m_i/V$ and $c_i = p_i/V$ respectively. According to Eqs. (1)–(3), the deterministic equations of the above variables become (see details in Methods)

$$\frac{dc_i}{dt} \approx \mu(\phi_i - c_i). \tag{7a}$$

$$\frac{dc_i^m}{dt} \approx \frac{1}{\tau}\left(k_0 \phi_i \phi_n \tau - c_i^m\right). \tag{7b}$$

A fixed point exists for the dynamics of $c_i$ and $c_i^m$, namely $c_i = \phi_i$ and $c_i^m = k_0 \phi_i \phi_n \tau$. This fixed point is stable due to the global nature of RNAPs and ribosomes: any noises arising from the copy number of RNAPs (ribosomes) equally affect all mRNAs (proteins), and therefore leave the relative fraction of one type of mRNA (protein) in the total pool of mRNAs (proteins) invariant. The average concentrations of mRNA and protein of gene $i$ become $\langle c_i \rangle = \phi_i$, and $\langle c_i^m \rangle = k_0 \tau \phi_i \phi_n$. The results are independent of the cell volume and genome copy number agreeing with experimental data on various organisms[18–20,22].

We take cell division explicitly into account and, for concreteness, use the "adder" model for cell division by considering an initiator protein $I$. The initiator protein accumulates from cell birth, triggers the cell division once it reaches the division threshold $I_c$ and is then destroyed (or "reset", e.g., after initiation of DNA replication in bacteria, the ATP-bound DnaA is dephosphorylated to the ADP-bound form)[46–48]. During a division event, we assume proteins and mRNAs are divided between the two daughter cells following a binomial distribution[49]. The initiator protein sets the scale of absolute protein number, and the average number of proteins produced in one cell cycle is equal to $\Delta(i) = I_c \phi_i / \phi_I$[47]. Since the protein number grows twofold during one cell cycle, the average protein number of gene $i$ at cell birth is $p_b(i) = I_c \phi_i / \phi_I$ and the corresponding average mRNA number at cell birth is $m_b(i) = k_0 I_c \tau \phi_i \phi_n / \phi_I$. We remark that the exact molecular mechanism of cell division does not affect our results.

We corroborate the above analytical calculations with numerical simulations. These will also capture the stochastic fluctuations in gene expression levels, which are not included in the previous analysis. Due to the short lifetime of mRNAs, the production of proteins can be approximated by instantaneous bursts[24]. We introduce the burst size parameter $b_0$ as the average number of proteins made per burst, $b_0 = k_t f_a \langle r(t) \rangle / \langle \sum_j m_j \rangle \times \tau \approx k_t f_a \phi_r / (k_0 \phi_n)$, independent of the cell volume. $\phi_i$ for $N = 200$ proteins are uniformly sampled in logarithmic space, with the sum over $\phi_i$ (including ribosome and RNAP) constraint to be precisely one. We choose the parameters to be biologically relevant for bacteria: the doubling time $T = \ln(2)/\mu = 150$ min, $r_b = 10^4$, $n_b = 10^3$, $b_0 = 0.8$, $I_c = 20$, $\phi_r = 0.2$, $f_a = 0.7$ and $\tau = 3.5$ min, see other numerical details in Methods. Our conclusions are independent of the specific choice of parameters.

In Fig. 2a, we show the typical trajectories from our simulations of cell volume, protein number and mRNA number for the same gene over multiple generations. To verify the exponential growth of protein and mRNA, we average the protein and mRNA numbers given a fixed relative phase in the cell cycle progression, which is normalized by the generation time and changes from 0 to 1. The averaged values of protein and mRNA numbers (circles) are well predicted by exponential growth, Eqs. (6a) and (6b) (black lines) without any fitting parameters, as shown in Fig. 2b with 3 single trajectories in the background. We also simulate a regulated gene with a time-dependent gene copy number and obtain qualitatively similar results (Methods, Supplementary Fig. 1).

The corresponding trajectories of protein and mRNA concentrations are shown in Fig. 2c, with bounded fluctuations around the predicted averaged values (black lines). In contrast, if the protein number and cell volume grow exponentially but independently, the ratio between them will diverge as the effects of noise accumulate, exhibiting a random walk behavior (Fig. 2d). Considering the cell cycle dependence of mRNA number and the homeostasis of protein concentration throughout the cell cycle, the experimental observation in *Escherichia coli* showing negligible correlations between mRNA number and protein concentration[50] is consistent with our model, and not contradictory to the strong correlation of mRNA concentration and protein concentration[51].

Within our model, we may also study the protein number dynamics: how does the protein number at cell division correlate with that at cell birth? We find that the correlations follow an "adder" (i.e. the number of new proteins added is uncorrelated with the number at birth), as shown in Fig. 2e. While this has been quantified in various organisms with respect to cell volume[8,9,11,52–54], checking correlations between protein content at cell birth and division has received significantly less attention[55,56]. Related to this, we study the auto-correlation function of protein concentration in time. We find that the auto-correlation function is approximately exponential, with a correlation time bounded from below by the doubling time (Supplementary Fig. 2). Both of these results provide experimentally testable predictions.

## Effects of finite duration of gene replication.

So far, we considered a constant $\phi_i$ throughout the cell cycle assuming an instantaneous replication of the genome. In this section, we relax this condition and study the effects of finite DNA replication time. We consider the bacterial model of DNA replication, specifically, *E. coli*, for which the mechanism of DNA replication is well characterized[57]. The duration of DNA replication is constant, and defined as the $C$ period. The corresponding cell division follows after an approximately constant duration known as the $D$ period. Details of the DNA replication model are in the Methods. In Fig. 3a, we show the time trajectories of the gene allocation fraction, mRNA concentration and protein concentration of one gene for a doubling time of $T = 30$ min with $C + D = 70$ min. The DNA replication introduces a cell cycle dependent modulation of $\phi_i$. The abrupt increase of $\phi_i$ corresponds to the replication of the specific gene $i$ (Fig. 3a) $\phi_i \to 2\phi_i$. However, as other genes are replicated, the relative fraction of gene $i$ in the total genome decreases. This modulation propagates to the mRNA concentration which essentially tracks the dynamics of $\phi_i$ due to its short lifetime. The modulation of mRNA concentration affects the protein concentration as well, yet with a much smaller amplitude. These results can be tested experimentally by monitoring the DNA replication process and mRNA concentration simultaneously.

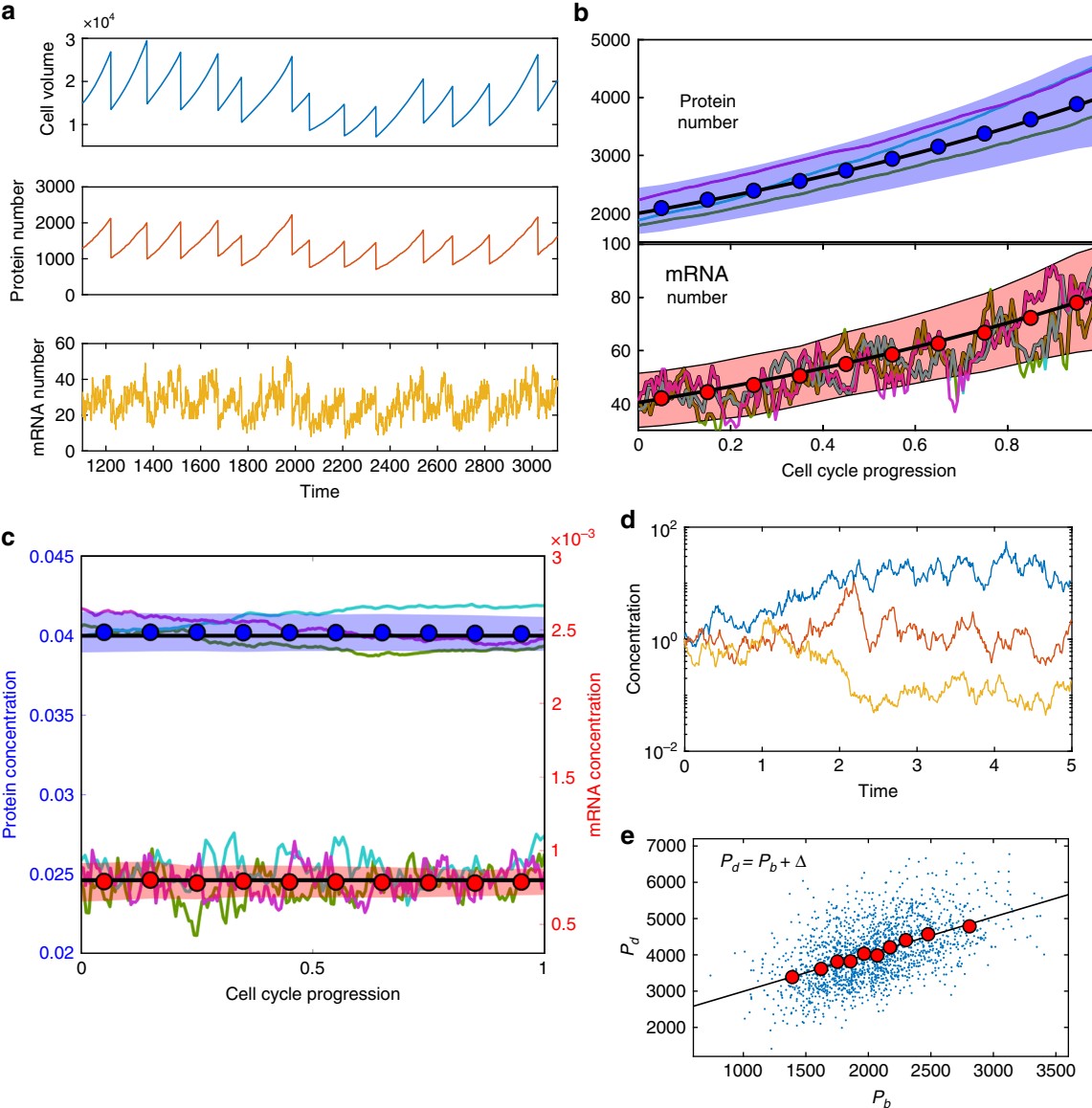

**Fig. 2** Exponential growth of the cell volume, protein number, mRNA number; the homeostasis of protein and mRNA concentrations throughout the cell cycle. **a** Numerical simulated trajectories of cell volume, protein number, and mRNA number are shown ($\phi_i = 0.018$). **b** The averaged values of protein and mRNA numbers of a highly expressed gene ($\phi_i = 0.04$), are shown (circles) with 3 single trajectories in the background. The black lines are theoretical predictions of Eqs. (6a) and (6b). The average is over 130 cell cycles. The color band represents the standard deviation (same for (**c**)). **c** The averaged values of protein and mRNA concentrations of the same gene as in (**b**) are shown (circles). The black lines are theoretical predictions of Eqs. (7a) and (7b). Three trajectories are shown in the background. **d** Three trajectories of diverging concentrations in the scenario where the protein number and cell volume grow independently. See the numerical details in Methods. **e** The scatter plot of the protein numbers at cell division ($P_d$) v.s. the protein numbers at cell birth ($P_b$). The circles are binned data. The black line is a linear fit of the binned data with slope 1.03, consistent with the adder correlations

Noise in gene expression can be classified as intrinsic and extrinsic noise[58]. While intrinsic noise is due to the stochastic nature of the chemical reactions involved in gene expression, extrinsic noise is believed to be due to the fluctuations of external conditions and common to a subset of proteins. Experiments have revealed a global extrinsic noise that affects all protein concentrations in the genome[50,59,60]. Because all genes are subjected to the finite duration of DNA replication, it is tempting to attribute the finite duration of DNA replication as one of the main sources of global extrinsic noise[34]. Within our model in the previous section (constant $\phi_i$'s throughout the cell cycle), there is no global extrinsic noise (Supplementary Fig. 3). A global extrinsic noise may emerge after we introduce the time-dependent $\phi_i$ due to DNA replication. However, we find that

the coefficient of variation (CV, the ratio between standard deviation and mean) of the most highly expressed proteins is only about 0.02 within the growing cell model (Fig. 3b), much smaller than that found in experiments[50,59]. We note that a small extrinsic noise due to gene replication is also observed in constant rate models[26,27]. Moreover, recent experiments and modeling have suggested that a significant part of the extrinsic noise of mRNA expression level can be attributed to the fluctuations of RNAP copy number[28]. Within our model, RNAP level fluctuations will lead to extrinsic noise in mRNA concentrations.

**A unified phase diagram of gene expression and cellular growth.** Experimental observations on *E. coli*[30] and budding

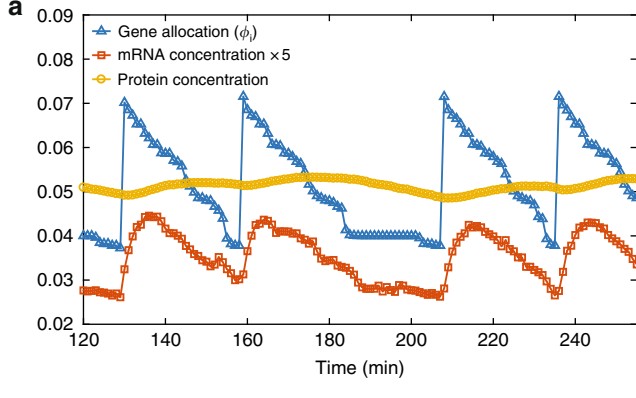

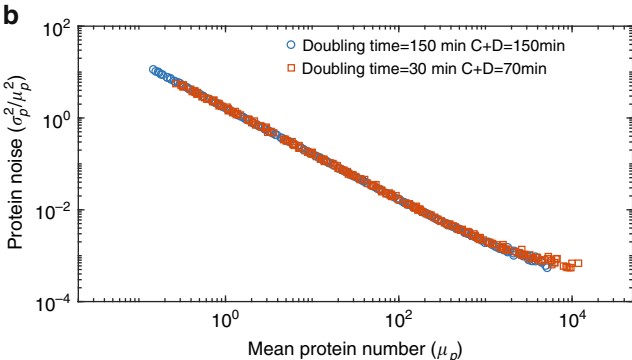

**Fig. 3** Effects of finite duration of DNA replication. **a** The time trajectory of gene allocation fraction (triangles), mRNA concentration (squares) and protein concentration (circles) of a high copy number protein ($\mu_p \approx 10^4$, see (**b**)). The doubling time is $T = 30$ min, and we use the values of the C and D periods from ref.[57], namely, $C = 35$ min and $D = 35$ min. In this situation, the cell undergoes DNA replication throughout the cell cycle. Nevertheless, the noise in $\phi_i$ does not propagate to the noise in protein concentration significantly. The value of mRNA concentration is 5 times amplified for clarity. **b** An exponentially growing population is simulated (See Methods). The noise magnitude is quantified as the square of CV of protein concentrations. The mean protein number ($\mu_p$) is the protein number per average cell volume. Gene dosage effects due to DNA replication do not generate a significant global extrinsic noise. Two different doubling times are considered

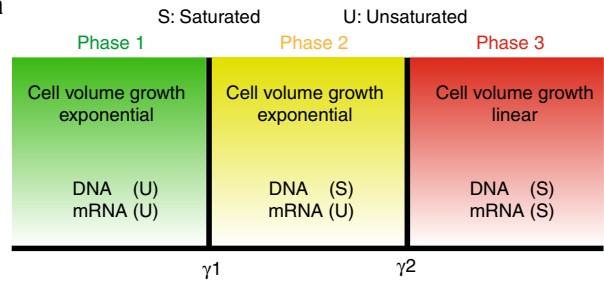

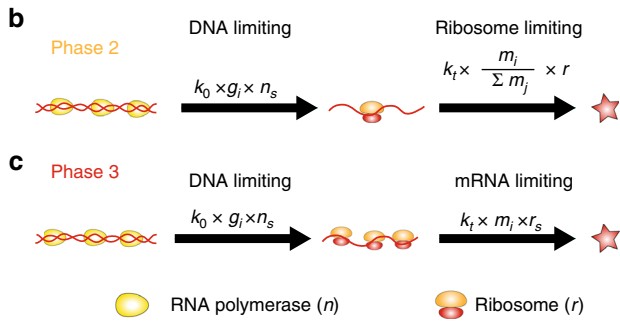

**Fig. 4** Phases of gene expression and cell volume growth. **a** Theoretical phase diagram of gene expression and cellular growth within our model. The x axis is the protein-to-DNA ratio ($\gamma$). When $\gamma < \gamma_1$, neither DNA nor mRNA is saturated. The mRNA number, the protein number and the cell volume all grow exponentially with the growth rate set by the fraction of ribosomal gene in the total genome ($\phi_r$). When $\gamma_1 < \gamma < \gamma_2$, DNA is saturated but mRNA is not. The protein number and the cell volume still grow exponentially while the mRNA number is a constant proportional to the gene number. When $\gamma > \gamma_2$, both DNA and mRNA are saturated. The protein number and cell volume grow linearly, and the cell volume growth rate is set by the genome copy number. **b** The gene expression dynamics in phase 2. In this phase, DNA is saturated by RNAPs, therefore, the transcription rate is proportional to the effective gene copy number, $g_i$. $n_s$ is the upper bound of the number of RNAPs that can work on one gene simultaneously. The translation rate is the same as in phase 1. To simplify the formula, we assume all ribosomes are active (to include the effect of an inactive fraction, $r$ should be replaced by $f_a r$). **c** The gene expression dynamics in phase 3, in which both DNA and mRNA are saturated. The translation rate is proportional to the mRNA number. $r_s$ is the upper bound of the number of ribosomes that can work on one mRNA simultaneously

yeast[32] support our assumption that ribosomes are limiting for translation. Experimental observations on plant and mammalian cells[18–20] and fission yeast[37] are also consistent with our assumption that RNA polymerase is limiting for transcription. However, as we discussed in the first section, in the same experiments on fission yeast[37] DNA became limiting for transcription at low DNA concentration. Therefore, we cannot exclude the possibility that in some cases because RNAPs are too abundant, DNA becomes the limiting resource for transcription rather than the number of RNAPs. Similarly, when ribosomes are too abundant relative to the transcript number, the limiting factor for translation becomes the transcript number rather than ribosome number.

In this section, we generalize our model by assuming that each gene has an upper bound on the number of RNAPs ($n_s$) than can simultaneously work on it. A possible extreme case is that the gene is fully loaded with RNAPs, on which RNAPs are only constrained by steric hindrance. The same assumption is made for mRNA with an upper bound of ribosomes ($r_s$) that can work on it simultaneously. We remark that the exact mechanism of DNA and mRNA saturation is beyond our coarse-grained model. If the number of RNAP (ribosome) is above the upper bound, the

transcription (translation) rate is limited by the gene (mRNA) number, in a similar fashion to the constant rate models.

We define the protein-to-DNA ratio (PTD ratio) as the sum of protein numbers divided by the sum of effective gene numbers,

$$\gamma = \sum_i p_i / \sum_i g_i. \qquad (8)$$

As the PTD ratio becomes larger, e.g., due to a sufficiently large cell volume with a fixed number of gene, the number of RNAPs (ribosomes) will exceed the maximum load the total genes (mRNAs) can hold. We have discussed thoroughly Phase 1 (neither DNA nor mRNA is saturated) earlier and we summarize our predictions on the transition from Phase 1 to other phases in the following.

Phase 2: In phase 2, the limiting factor in transcription becomes the gene copy number and the transcription rate is proportional to the gene copy number (Fig. 4b). The threshold PTD ratio for the transition from Phase 1 to Phase 2 is

(Methods),

$$\gamma_1 = \frac{n_s}{\phi_n}. \tag{9}$$

Here $n_s$ is the upper bound of the number of RNAPs that can work on one gene and $\phi_n$ is the gene allocation fraction of RNAP. Because mRNA is not saturated, the protein number and the cell volume grow exponentially with the same growth rate as Phase 1, Eq. (5), and the homeostasis of protein concentration is still valid. However, because the production rate of mRNA is now proportional to the gene copy number, the mRNA concentration is not constant anymore as the cell volume grows (Methods). In Phase 2, even though the transcription rate doubles after the genome is replicated, the translation rate is proportional to the relative fraction of mRNA in the total pool of mRNAs and therefore still independent of the genome copy number. The average protein concentration is equal to the gene allocation fraction ($\langle c_i \rangle = \phi_i$). Recent proposed theoretical models of gene expression are consistent with this phase[61]. In terms of transcription, our model in Phase 2 is equivalent to constant rate models and we have confirmed that for both bacteria and mammalian cells, the typical lifetime of mRNA is short enough compared with the doubling time to distinguish Phase 1 and Phase 2 (Supplementary Fig. 4).

Phase 3: As the cell volume becomes larger, mRNA may get saturated as well. The limiting factor for translation is now the mRNA copy number (Fig. 4c). The threshold PTD ratio for the transition from Phase 2 to Phase 3 is (Methods)

$$\gamma_2 = \frac{k_0 \tau r_s n_s}{\phi_r}. \tag{10}$$

Here $r_s$ is the upper bound of the number of ribosomes that can work on one mRNA. In this phase, the translation rate is proportional to the mRNA number and the protein number grows linearly as $\dot{p}_i = k_t k_0 g_i \tau n_s r_s$, with a linear growth rate proportional to the gene number. Therefore, within the assumption that the cell volume is determined by the total protein number, the cell volume grows linearly as well with the linear growth rate proportional to the total gene number,

$$\mu_l = k_t k_0 \tau n_s r_s \sum_i g_i, \tag{11}$$

and therefore proportional to the genome copy number. As in Phase 2, the mRNA concentration decreases as the cell volume grows, however, the protein concentration is still constant with the average protein concentration equal to the gene allocation fraction ($\langle c_i \rangle = \phi_i$, Methods). In Phase 3, even though the cell volume grows linearly, the population still grows exponentially with a population growth rate. However, there is no general relation between the ribosomal fraction in the proteome and the population growth rate, in contrast to the growth law in Phase 1 and 2. We summarize the predicted phase diagram of cellular growth in Fig. 4a.

To gain some sense regarding the parameters associated with our proposed phase diagram, we estimate the PTD ratio of E. coli. Considering the typical cell volume of E. coli as 1 μm³, the protein density as $3 \times 10^6$ proteins/μm³ and the total number of protein-coding genes in E. coli as 4000[62], we estimate the protein-to-DNA ratio for E. coli as $\gamma \sim 1000$. Estimates of the two threshold values of PTD ratios (see Methods) suggest that $\gamma_1 \sim 1500$ and $\gamma_2 \sim 20,000$.

These estimates suggest that wild-type E. coli cells are found in Phase 1, but close to Phase 2. We remark that the actual threshold values of PTD ratio for the transitions between different growth phases may be affected by other factors, e.g., the heterogeneous

size of genes, but we propose that the general scenario of the transition from Phase 1 to Phase 3 as the protein-to-DNA ratio increases should be generally applicable. As the PTD ratio increases, we predict a transition from exponential growth to linear growth for protein number and cell volume (Supplementary Fig. 5). We propose future experiments to study the potential transition from exponential to linear growth of cell volume, for example using filamentous E. coli where cell division and gene replication are inhibited. Similar experiments can also be done for larger cells, e.g., mammalian cells, in which the transition from exponential growth to linear growth of cell volume may be easier to achieve. Preliminary results from experiments measuring the growth of cell mass of mammalian cells[63] and yeast cells[64] indeed show a crossover from exponential growth to linear growth when the cell mass is above a threshold value, consistent with our prediction.

It has been shown in bacteria that there are excess RNAPs nonspecifically bound to DNA[40,41]. In the Methods, we consider a modified model taking into account the partitioning of RNAPs to free RNAPs, elongating RNAPs, promoter-bound RNAPs and nonspecifically bound RNAPs. The transcription rate is determined by the concentration of free RNAPs through Michaelis-Menten kinetics[40,65]. We find that our conclusions remain intact with an approximately constant fraction of actively transcribing RNAPs in the total RNAPs for Phase 1 (Supplementary Fig. 6). The effect of nonspecifically bound RNAPs is therefore to renormalize the transcription constant $k_0$ in Phase 1 (Eq. (1)) by a constant factor. The transition from Phase 1 to Phase 2 is qualitatively unaffected (Supplementary Fig. 7) and the threshold PTD ratio $\gamma_1$ (Eq. (9)) from Phase 1 to Phase 2 is changed by a constant factor (Methods). We note that alternative mechanisms of gene saturation can occur upon introducing the different classes of RNAPs, through the saturation of free RNAPs and the Michaelis-Menten kinetics (Methods).

## Discussion

In this work, we propose a coarse-grained model of stochastic gene expression incorporating cell volume growth and cell division. In the first part, we consider the biological scenario that RNAPs are limiting for transcription and ribosomes are limiting for translation. In other words, neither DNA nor mRNA is saturated. We find that the limiting nature of ribosomes in the translation process leads to the exponential growth of protein numbers. The limiting nature of RNA polymerase and its exponential growth lead to the exponential growth of mRNA numbers. Homeostasis of protein concentrations originates from the fact that ribosomes make all proteins. Homeostasis of mRNA concentration comes from the resulting bounded concentration of RNAPs. Our model is consistent with the constancy of mRNA and protein concentration as the genome copy number varies since the transcription rate depends on the relative fraction of genes in the genome rather than its absolute number[22].

During DNA replication, we find that the gene allocation fraction $\phi_i$ for one specific gene doubles after the gene is replicated but decreases afterwards since other genes are replicated as well and compete for RNAPs. This prediction can be tested by monitoring the mRNA concentration and the copy number of one gene throughout the cell cycle. Furthermore, we extend our model to more general cases in which DNA and mRNA can be saturated by an excess of RNAP and ribosome. We find three possible phases of cellular growth as the protein-to-DNA ratio $\gamma$ increases. A transition from exponential growth to linear growth of protein number and cell volume is predicted. In the future, it will be interesting to study the interplay between the global interactions which are the focus of this work and local

interactions between genes. Our model provides an alternative model to constant rate models to study genetic networks, which would be advantageous when cell cycle effects are important.

## Methods

**Derivation of protein and mRNA concentrations.** We define the fraction of mRNA $i$ in the total mRNA pool as $f_i = m_i / \sum_j m_j$, and the concentration of mRNA and protein of gene $i$ as $c_i^m = m_i/V$, $c_i = p_i/V$. We denote the RNAP and ribosome concentrations as $c_n$ and $c_r$, respectively. According to Eqs. (1)–(3), the deterministic equations of the above variables then become

$$\frac{df_i}{dt} = \frac{k_0 n}{\sum_j m_j}(\phi_i - f_i) \qquad (12)$$

$$\frac{dc_i}{dt} = k_t c_r f_a (f_i - c_i) \approx \mu(f_i - c_i). \qquad (13)$$

$$\frac{dc_i^m}{dt} = \frac{1}{\tau}\left(k_0 \phi_i c_n \tau - (1 + \mu\tau)c_i^m\right). \qquad (14)$$

Using the condition that mRNA degradation time is much smaller than the doubling time ($\mu\tau \ll 1$), we find the fixed points for the dynamics of $f_i$, $c_i$, and $c_i^m$. These are, $f_i = c_i = \phi_i$ and $c_i^m = k_0 \phi_i c_n \tau$. Replacing $f_i$ by $\phi_i$ and $c_n$ by $\phi_n$, we obtain the approximate version of the above equations, Eqs. (7a) and (7b).

**Simulations of independent growth model.** In the growth model corresponding to Fig. 2d, we assume the protein number and cell volume grow exponentially and independently,

$$\frac{dp}{dt} = \left(1 + \xi_p(t)\right)p \qquad (15)$$

$$\frac{dV}{dt} = (1 + \xi_V(t))V. \qquad (16)$$

Here, $\xi_p(t)$, $\xi_V(t)$ are white noise terms, with the auto-correlation function, $\left\langle \xi_{p,V}(0)\xi_{p,V}(t) \right\rangle = A_{p,V}\delta(t)$. In Fig. 2d of the main text, we choose $A_p = A_V = 1$.

**Simulations of growing cell model.** We simulated Eqs. (1)–(3), fixing $r_b$, $n_b$, $b_0$, $\phi_r$, $f_a$, $I_c$, $\tau$ as well as the growth rate $\mu$. Other parameters are inferred given the above values, e.g., $\phi_n = n_b\phi_r/r_0$, $k_t = \mu/(\phi_r f_a)$, $k_0 = k_t f_a r_b/(b_0 n_b)$. We fix the time step $\delta t$ so that the probability for one event to happen during a time step is smaller than 0.1. We track one of the daughter cells after cell division.

**Gene dosage effects.** In reality, the gene allocation fraction $\phi_i$ changes during the cell cycle due to the finite duration of DNA replication. In this section we introduce the modified version of the gene expression model incorporating DNA replication. Although our model is general, we focus on DNA replication in bacteria for concreteness, specifically *E. coli* where this process is very well characterized. We expect our conclusions to be generally valid. Furthermore, we refine our model for cell division, assuming that the initiator protein triggers the initiation of DNA replication rather than cell division, with the threshold $I_c$ proportional to the number of origins of replication[57,66] (the number of which doubles at each initiation). We assume that the cell division takes place a fixed time $C + D$ after initiation of the DNA replication, where $C$, $D$ are respectively the time for DNA replication and the time between the completion of DNA replication and cell division. The number of origins reduce by half at each cell division. Other details are the same as in the main text. Each gene doubles its copy number during the $C$ period, and we choose this gene replication time to be randomly and uniformly distributed across all genes. When a gene $i$ replicates,

$$\phi_i \rightarrow 2\phi_i \qquad (17)$$

$$\phi_j \rightarrow \frac{\phi_j}{\sum_k \phi_k}, \qquad (18)$$

where the second equation accounts for the normalization of the gene allocation fraction. We choose the experimentally reported $C$ and $D$ and cell doubling time from ref.[57]. In Fig. 3a, we simulate the model by tracking one daughter cells. In Fig. 3b, we track all the cells in an exponentially growing population, which starts from 100 cells to 5000 cells.

**Simulations of gene activation.** We generalize the constitutive expressed genes considered in the main text to include a single regulated gene by considering a

random telegraph process of the effective gene copy number[1],

$$g_{i0} \underset{k_g^+(c_{TF})}{\overset{k_g^-}{\rightleftharpoons}} 0. \qquad (19)$$

Here the gene deactivation rate $k_g^-$ is constant, and the activation rate is set by the concentration of transcription factor through positive regulation, $k_g^+ = k_{g0}c_{TF}$. Here, $k_{g0}$ is constant. When gene $i$ is active, the corresponding gene allocation fraction follows $\phi_i = g_{i0}/\sum_j g_j$, and when it becomes deactivated $\phi_i = 0$. Note that here we only consider one regulated gene $i$, but the changing gene allocation of gene $i$ also affects other genes' allocation fraction. We simulate the model in Phase 1, and the deactivation of gene $i$ increases other genes' allocation fraction as $\phi_j \rightarrow \phi_j/(1 - \phi_i)$, with $\phi_i = g_{i0}/\sum_j g_j$.

Simulated trajectories of gene allocation fraction, mRNA number, protein number and cell volume are shown in Supplementary Fig. 1.

**General model of gene expression.** We consider the generalized equation of mRNA number, Eq. (1) in the deterministic limit as

$$\dot{m}_i = \begin{cases} k_0 \phi_i n - m_i/\tau, & \text{if } n < n_c, \\ k_0 g_i n_s - m_i/\tau, & \text{if } n \geq n_c. \end{cases} \qquad (20)$$

Here $n_c$ is the threshold number of RNAPs above which DNA starts to be saturated, in which case the transcription rate becomes proportional to the effective gene copy number $g_i$ and independent of the RNAP number. For one gene, the maximum load of RNAP that it can hold is $g_i n_s$, where $n_s$ is the maximum number of RNAPs that a single copy of constitutively expressed gene ($g_i = 1$) can hold. $n_c$ can be computed as

$$\phi_i n_c = g_i n_s \Rightarrow n_c = \sum_i g_i n_s. \qquad (21)$$

We also generalize the growth of protein number from Eq. (3) to

$$\dot{p}_i = \begin{cases} k_t \frac{m_i}{\sum_j m_j}r, & \text{if } r < r_c, \\ k_t m_i r_s, & \text{if } r \geq r_c. \end{cases} \qquad (22)$$

Here $r_c$ is the maximum number of ribosomes above which mRNA starts to be saturated. We drop the fraction of actively working ribosomes since it is often a constant depending on the growth condition[30]. $r_s$ is the maximum number of ribosomes one mRNA can hold. We can calculate $r_c$ as

$$\frac{m_i}{\sum_j m_j}r_c = m_i r_s \Rightarrow r_c = \begin{cases} k_0 \tau n r_s, & \text{if } n < n_c \\ k_0 \tau n_c r_s, & \text{if } n \geq n_c \end{cases} \qquad (23)$$

Given Eqs. (20) and (22), we obtain four possible phases: (i) $n < n_c$, $r < r_c$, (ii) $n > n_c$, $r < r_c$ (iii) $n > n_c$, $r > r_c$ and (iv) $n < n_c$, $r > r_c$. Given a fixed value of $\phi_r$ and $\phi_n$, either (ii) or (iv) is possible. Realization of (ii) requires that $n > \sum_i g_i n_s$ and $r < k_0 \tau r_s \sum_i g_i n_s$, therefore

$$\frac{\phi_n}{\phi_r} > \frac{1}{k_0 \tau r_s}. \qquad (24)$$

In cases where Eq. (24) breaks down, a finite fraction of ribosomes are not utilized. This requires a large fraction of genes in the genome making ribosomes that cannot translate because mRNAs are saturated. Since ribosomes are typically more expensive to make than other proteins[30,33], we assume the biological scenario, Eq. (24) will be satisfied. From Eq. (21) and using $n/\sum_i p_i = \phi_n$, we obtain the threshold PTD ratio for the transition from Phase 1 to Phase 2,

$$\gamma_1 = \frac{n_s}{\phi_n}. \qquad (25)$$

In Phase 2, the average mRNA concentration becomes

$$\langle c_i^m \rangle = \frac{k_0 g_i n_s \tau}{V} = \frac{k_0 \phi_i n_s \tau \sum_i g_i}{V} = \frac{k_0 \phi_i n_s \tau}{\gamma}, \qquad (26)$$

which is inversely proportional to the protein-to-DNA ratio.

From Eq. (23) and using $r/\sum_i p_i = \phi_r$, we obtain the transition PTD ratio from Phase 2 to Phase 3 as,

$$\gamma_2 = \frac{k_0 \tau r_s n_s}{\phi_r}. \qquad (27)$$

In Phase 3, the mRNA concentration is the same as Phase 2. Because the cell volume is the sum of all proteins, the protein concentration is the same as Phase 2 and Phase 1, $\langle c_i \rangle = g_i/\sum_i g_i = \phi_i$.

**Estimation of the threshold protein-to-DNA ratios for *E. coli*.** We approximate the upper bound of RNAP number working on a single gene as roughly equal to the number of RNAPs on a typical gene (~$10^3$ base pairs) when half of the gene is occupied. The linear size of RNAP is about 5 nm, and the length of one base pair is about 0.3 nm, leading to the estimate $n_s \sim 30$. A similar calculation for the upper bound of ribosome on a single mRNA leads to $r_s \sim 10$ since ribosome's linear size is about 3 times larger than RNAP[62].

We take $\phi_r \approx 0.2$ according to the ref.[30], and estimate the gene allocation fraction of RNAP to be $\phi_n \sim 0.02$ since the number of RNAPs in *E. coli* is roughly 10% of the number of ribosomes[62]. We estimate the life time of mRNA as 5 min[62].

We estimate the transcription rate of one RNAP by considering two potential limiting steps in transcription and taking the slower one. First, assuming the initiation of transcription is diffusion limited, we could estimate the time scale for one RNAP to bind the transcription site as $\Delta t \sim 1\ \mu m^2/(0.2\ \mu m^2/s) \sim 5\ s$ using the measured diffusion constant of RNAP[41,67]. Second, we could also estimate the elongation time as the typical length of gene divided by the elongation rate of RNAP, $\Delta t \sim 1000\ nt/50(nt/s) \sim 20\ s$[62]. Taking the slower time scale from the above two calculations, we estimate $k_0 \approx 0.05\ s^{-1}$. Finally, we compute $\gamma_1$ and $\gamma_2$ using the above estimated parameters, and obtain $\gamma_1 \sim 1500$, $\gamma_2 \sim 20,000$.

**Effect of nonspecifically bound RNAPs.** Previous studies on bacteria have shown that there are excess RNAPs bound nonspecifically to the genome and modeled their kinetics[40,41]. In this section, we consider a modified model to take into account nonspecifically bound RNAPs. For our purpose, we consider a simplified model with four classes of RNAPs, namely, (i) elongating RNAPs, $n_e$ (ii) RNAPs bound to a promoter, $n_p$ (iii) RNAPs nonspecifically bound to DNA, $n_{ns}$ (iv) free RNAPs, $n_{free}$. We assume a Michaelis-Menten relation for the number of promoter-bound RNAPs and nonspecifically bound RNAPs[40,65],

$$n_p = G \frac{c_{free}}{c_{free} + K_s}, \quad (28)$$

$$n_{ns} = G g_{ns} \frac{c_{free}}{c_{free} + K_{ns}}. \quad (29)$$

Here $c_{free}$ is the concentration of free RNAPs and $K_s$, $K_{ns}$ are the Michaelis constants. $G$ is the total number of genes and $g_{ns}$ is the number of nonspecific binding sites per gene. Note that $c_{free} = c_n F_{free}$, with $c_n$ the concentration of total RNAPs and $F_{free}$ the fraction of free RNAPs in the total RNAP pool. For simplicity, we assume one promoter for each gene.

The number of elongating RNAPs is then given by:

$$n_e = n_p k_{cat} \tau_0 = n_p \Lambda, \quad (30)$$

where $k_{cat}$ is the transition rate from promoter-bound RNAPs to elongating RNAPs, $\tau_0$ is the time for transcribing a gene, and $\Lambda = k_{cat}\tau_0$ is the ratio between the number of elongating RNAPs and promoter-bound RNAPs[40]. We consider parameter regimes motivated by typical biological scenarios, in particular (1) the number of sites for nonspecific binding is much larger than the number of promoters, (2) nonspecific binding of RNAPs to DNA is much weaker than the specific binding of RNAPs to promoters, (3) the number of promoter-bound RNAPs is small compared with elongating RNAPs.

Using $n = n_e + n_p + n_{ns} + n_{free}$, we can find a self-consistent equation for $F_{free}$,

$$c_n\gamma\left(1 - F_{free}\right) = (1+\Lambda)\frac{c_n F_{free}}{c_n F_{free} + K_s} + g_{ns}\frac{c_n F_{free}}{c_n F_{free} + K_{ns}}. \quad (31)$$

Here $\gamma$ is the protein-to-DNA (PTD) ratio, i.e., the ratio between the total number of proteins (equivalent to cell volume $V$ within our model) and the total gene number, $\gamma = V/G$. We can use Eq. (31) to compute the fraction of free RNAPs given a PTD ratio and use Eqs. (28)–(30) to compute the fraction of elongating RNAPs, $F_e$ and nonspecifically bound RNAPs, $F_{ns}$.

Since the left side of Eq. (31) monotonically decreases from $c_n\gamma$ to 0 as $F_{free}$ increases from 0 to 1 and the right side of Eq. (31) monotonically increases as $F_{free}$ increases, we find that as the protein-to-DNA ratio increases, $F_{free}$ increases. Previous studies have shown that $F_{free} \approx 0.1$ for bacteria[40,41] and support the assumption that $c_{free}$ is smaller or comparable to $K_s$[40,65] (note that nonspecific binding is characterized by $K_{ns}$ larger than $K_s$). In the following, we first assume a small $F_{free}$ and $c_{free} \ll K_s$, and show this leads to a behavior qualitatively equivalent to Phase 1 of the main text, albeit with a renormalization of the transcription constant $k_0$. We also later discuss the situations when these conditions break down, and show that they lead to behavior consistent with Phase 2. The transcription rate for one specific gene with an effective gene copy number $g_i$ can be written as

$$TR_i = \frac{1}{\tau_0} g_i \frac{n_e}{G} = \frac{1}{\tau_0} \phi_i n F_e \quad (32)$$

Here $g_i/G = \phi_i$ is the gene allocation fraction of gene $i$, and $F_e$ is the fraction of elongating RNAPs in the total RNAP pool. In the limit of a small $F_{free}$ and

$c_{free} \ll K_s$, $F_e$ becomes a constant:

$$F_e \approx \frac{n_e}{n_e + n_p + n_{ns}} = \frac{\Lambda}{1 + \Lambda + g_{ns}K_s/K_{ns}}. \quad (33)$$

Therefore, the transcription rate corresponds to Phase 1 of our model, $TR_i = \tilde{k}_0 \times \phi_i \times n$ with $\tilde{k}_0 = k_0 F_e$.

In the limit that all RNAPs are actively transcribing and no nonspecifically bound RNAPs, $\Lambda \gg 1$ and $g_{ns} = 0$, we go back to the situation that all RNAPS are working with $\tilde{k}_0 = 1/\tau_0 = k_0$. Therefore we conclude that the introduction of nonspecifically bound RNAPs does not affect our model qualitatively, and its effect is to renormalize the transcription constant $k_0$ in Eq. (1) by a constant factor, $\tilde{k}_0 = k_0 F_e$.

We simulate a single lineage of growing cells using the full model (with partitioning of RNAPs and gene replication). We set the parameters as: $\Lambda = 50$, $g_{ns} = 1000$, $k_0 = 1$ min$^{-1}$, $G = 2000$ (before gene duplication), $K_s = 0.02\langle c\rangle$, $K_{ns} = 0.8\langle c\rangle$, where $\langle c\rangle$ is the total protein concentration within the cell which we set to 1 throughout the paper. Our conclusions are independent of the specific values of parameters. The gene allocation fractions are the same as the main text and the average RNAP concentration during the cell cycle $\langle c_n\rangle = \phi_n = 0.02$. The fractions of elongating RNAPs and nonspecifically bound RNAPs are approximately constant with a small cell cycle modulation (the coefficient of variation is of the order of 0.01), consistent with the above results since $F_{free}$ is small (Supplementary Fig. 6a). We also find a linear scaling between mRNA number and cell volume, consistent with Phase 1 of our model (Supplementary Fig. 6b).

We next consider the transition from Phase 1 (RNAP limiting) to Phase 2 (gene limiting). Assuming the saturation of genes is due to the steric hindrance of elongating RNAPs with a minimum distance between two successive RNAPs, we can find the threshold PTD ratio from Phase 1 to Phase 2. Since the fraction of elongating RNAPs is constant, we can compute the threshold PTD ratio using $nF_e/G = n_s$ (where $n_s$ is the upper bound of the number of RNAPs that can work on one gene simultaneously), and obtain

$$\bar{\gamma}_1 = \frac{n_s}{\phi_n}\frac{1 + \Lambda + g_{ns}K_s/K_{ns}}{\Lambda}. \quad (34)$$

We find that the introduction of nonspecifically bound RNAPs does not affect our main results qualitatively and it changes the threshold PTD ratio from Phase 1 to Phase 2, Eq. (9), by a constant factor, which is the inverse of the fraction of elongating RNAPs.

In Supplementary Fig. 7, we show the transition from Phase 1 to Phase 2 in the proposed experiment in which cell division is blocked, in the presence of nonspecific binding. At each point in time we solve the self-consistent equation of the partitioning of RNAPs, Eq. (31). The plot shows both the deterministic dynamics of average mRNA number (black lines) as well as the results of the stochastic simulations (red/blue solid lines). We compare the dynamics for two cells with different fixed genome sizes. Initially, the two cells are in Phase 1 and therefore have identical mRNA number proportional to cell volume. When they exceed their respective threshold PTD ratio, the mRNA number begins to saturate and becomes limited by the gene number, therefore the cell with a twice larger genome size has twice more mRNAs (Phase 2).

So far we assumed that $F_{free}$ is small and $c_{free}$ is smaller than the Michaelis constant. In the following we relax these assumptions. We find that the introduction of the different RNAP classes introduces alternative mechanisms for gene saturation, that lead to behavior consistent with Phase 2.

We first relax the assumption that $F_{free} \ll 1$ (assuming that $c_{free} \ll K_s$ still holds). Because the transcription rate for a specific gene is

$$TR_i = \frac{1}{\tau_0} g_i \Lambda \frac{c_n F_{free}}{K_s}, \quad (35)$$

when $F_{free}$ is comparable to 1, the transcription rate will be saturated as well and proportional to the gene copy number $g_i$. From Eq. (31), we find that $F_{free} \approx \gamma/\left(\frac{1+\Lambda}{K_s} + \frac{g_{ns}}{K_{ns}} + \gamma\right)$, therefore the threshold PTD ratio for $F_{free}$ to be comparable to 1 is

$$\gamma_{1,F} = \frac{1+\Lambda}{K_s} + \frac{g_{ns}}{K_{ns}}. \quad (36)$$

Second, we can relax the assumption that $c_{free} \ll K_s$ (assuming $F_{free} \ll 1$ still holds). When $c_{free} \approx K_s$, the transcription rate can be saturated as well. Using $F_{free} \approx \gamma/\left(\frac{1+\Lambda}{K_s} + \frac{g_{ns}}{K_{ns}}\right)$, we can obtain the corresponding threshold PTD ratio as

$$\gamma_{1,M} = \frac{1 + \Lambda + g_{ns}K_s/K_{ns}}{\phi_n}. \quad (37)$$

We remark that the particular mechanism which drives the cell to Phase 2 (gene limiting) is the one with the smallest threshold. Comparison between Eqs. (34),

(36) and (37) shows that when $n_s/\phi_n < \Lambda/K_s$ and $n_s < \Lambda$, genes get saturated due to steric hindrance.

Reference[68] shows that for wild type E. coli in fast growth conditions the mRNA levels in the cell do not change when the DNA amount is lowered. Within our model this is consistent with Phase 1, and inconsistent with Phase 2, thus suggesting that $F_{free} \ll 1$ and $c_{free} \ll K_s$ as discussed earlier.

## Data availability

The data that support the findings of this study are available from the authors on request.

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

## Acknowledgements

We thank Naama Barkai, Ido Golding, Andreas Hilfinger, Po-Yi Ho, Meriem El Karoui, Andrew Murray, Johan Paulsson, Leonardo A. Sepúlveda, and Sven van Teeffelen for useful discussions related to this work. AA thanks the A.P. Sloan foundation, the Milton Fund, the Volkswagen Foundation and Harvard Dean's Competitive Fund for Promising Scholarship for their support. JL was supported by the George F. Carrier fellowship and the National Science Foundation through the Harvard Materials Research Science and Engineering Center (DMR-1420570).

## Author contributions

All authors conceived the work, carried out the work, and jointly wrote the manuscript.

## Additional information

**Competing interests:** The authors declare no competing interests.

