## [Peer Review File · Nature Communications]

Reviewers' comments:

Reviewer #1 (Remarks to the Author):

The authors introduce a new course-grained model of stochastic gene expression, that seeks to explain the homeostasis of protein and mRNA concentrations in exponentially growing cells. This cannot be done with standard current gene expression models, which generally assume a fixed cell volume with constant transcription and translation rates that are proportional to the gene copy number. Based on an analysis of this model, the authors make a number of predictions about the mechanisms underlying the observed homeostasis in protein and mRNA concentrations. The results presented clearly make a contribution to the literature on modelling of stochastic gene expression. Whether these results are sufficiently clearly explained or important to warrant publication in an interdisciplinary journal like Nature Communications I am unsure.

Major comments:

1. The authors do a rather poor job of explaining the contribution and importance of their work to readers outside the particular field of stochastic gene expression modeling. Many terms are used without being clearly defined (e.g. intrinsic versus extrinsic noise) and sentences such as "The homeostasis of protein and mRNA concentrations imply that there must be a regulatory mechanism in place to prevent the accumulation of noise over time and to maintain a bounded distribution of concentrations" are unnecessarily complicated and assume a fair degree of specialist knowledge on the part of the reader – what "noise" specifically could be being accumulated over time here, and what are the "bounded distributions" referred to (this is the first time such an expression is used in the paper)? I think the authors need to "zoom out" a little and begin the paper with a much more readable explanation of what they are trying to do.
2. The authors develop a simple model and simulate / analyse it to make some predictions. As such, I am uncomfortable with their repeated use of the term "we show that". There is no data in the paper – the results are simply hypotheses (hopefully testable ones – this is not well discussed) that derive from the assumptions on which the model is based. If one agrees that the assumptions are reasonable, then the resulting predictions should be plausible, but they will only be "shown" to be true if/when they are confirmed experimentally (would the authors not agree?). It would be more accurate to use the term "our model predicts that...". Further the claim that "We verify our theoretical results using numerical simulations" is not accurate – numerical simulations simply implement theoretical (mathematical) equations on a computer, they do not "verify" anything. In general, the paper needs a much more convincing discussion about how the various predictions made by the model could be tested and confirmed experimentally.
3. The limitations of the proposed modeling approach need to be more clearly discussed. The authors consider only the case of constitutively expressed genes, where the only interactions between said genes are the competitions for ribosomes and RNAPs. This is a rather limited case, and it is not really clear whether the proposed approach can be extended to more realistic / complex scenarios. Other than for the purposes of investigating the specific questions considered in this paper, is there any particular advantage to the author's proposed modeling framework, over standard fixed-volume models (which certainly can take more complex interactions in to account)? If there is, it is not clear to me at the moment.
4. The authors should discuss the implications of the results in the following paper for their work, as it seems to be closely related: <https://www.biorxiv.org/content/early/2018/02/18/267658>

Reviewer #2 (Remarks to the Author):

This paper addresses several questions related to gene expression experiments, in particular, the gene copy number and cell cycle dependence of mRNA and protein concentrations, the source of the extrinsic noise observed in the Taniguchi et al study (ref. 28), and the missing correlation between mRNA and protein seen in the same study. All these questions (possibly with the exception of the last one) are timely and of some importance, so a study of this type is very welcome. Nevertheless, I am not entirely convinced the importance of the paper is high enough to justify publication in a high-impact journal. The scope of what the authors try to address results is very wide, which has the unfortunate consequence that some issues that could have been nicely analyzed separately are not clearly distinguished and also previous work is not always recognized.

Overall, I find the first aspect, the gene copy number and cell cycle dependence the most intriguing of the 3 topics of the paper. The extrinsic noise part remains somewhat inconclusive, although plausible; and I do not see the point of the last part about correlations (basically, this is a remark that earlier work was flawed in a rather obvious way).

Here are a few more detailed comments:

1) Volume growth and cell cycle: The authors study two aspects of gene expression here that are often not included in simpler models: the growth of the cell volume and the competition of genes for machinery. They refer to previously models without the two as fixed-volume models. In my opinion, this is not correct, as some previous models have studied the first aspect (e.g. Jones et al, ref. 31, but also two papers by Bierbaum and Klumpp, J Stat Phys 2012 and Phys Biol 2015.) One result of these studies is that the protein concentration does not vary strongly over the cell cycle, but the mRNA concentration does, which is related to different half-lives. In the Jones et al. paper the predicted variation seems to agree with experimental data, although they do not measure time series of the mRNA concentration directly, which would be needed for a definite answer.

What is really new in this paper is that competition for machinery is also included, which in my understanding suppresses the small but systematic variations over the cell cycle that were seen in the earlier papers. For mRNA this would be a testable prediction where the models differ.

2) I find the language used by the authors sometimes potentially misleading. They state that the number of ribosomes rather than the number of mRNA limits translation (and correspondingly for mRNA). In my view, this is true on a global level but not for individual genes (and I think the authors agree with that, as this is what they formulate in their model). This distinction is very important and I think this should be phrased more clearly.

3) While there is general agreement that ribosomes are limiting for translation in the sense that (almost) all ribosomes are active in translation and mRNAs compete for them, I am less convinced the same is true for RNA polymerases and genes. At least for E. coli, it has also been claimed that there is an excess of RNA polymerases (e.g. the model by Klumpp & Hwa 2008 and the experiments by Bakshi et al 2013) . Given that protein concentrations become almost constant even with a gene copy number dependence of transcription, again mRNA should be the quantity to look at here. If a more general model lies somewhere between no competition as in the studies mentioned above and the perfect competition studied here, it would be interesting to see how big the effect of competition is.

4) Noise in ribosomal gene allocation: Here I tend to find the model plausible, but likely not a unique solution. Yes, this could contribute to the observed extrinsic noise, but couldn't other factors do that too? There is evidence for fluctuations in metabolism and distributions of growth rate. Why should ribosomes be the dominant factor here?

5) mRNA-protein correlation: In my opinion, this discussion is quite artificial. It starts with the fact that in the Taniguchi et al paper (ref. 28), mRNA and protein copy numbers are compared. Their averages should be correlated if different conditions are considered, but the instantaneous numbers in individual cells should not be strongly correlated, since the two quantities have very different time scales. I think that the Taniguchi et al. paper (which other than that is an outstanding landmark study) has contributed to confusion as it suggests that this is a surprising observation, while at the same time it also demonstrates that not. Ref 29 discusses the correlations from ref. 28 further and basically shows that these low correlations are not consistent with a large class of stochastic models. The authors of the paper under review claim now that ref. 29 correlates numbers of mRNA (number per cell, i.e. variable volume) and protein concentration (number per constant volume) rather than numbers in constant volume as required by their relation. I have not checked that, but if this criticism of ref. 29 is correct and it looks like it is, then the analysis of ref. 29 is simply wrong and what the authors describe as a puzzle in the field does not exist. I am wondering whether a short comment to the journal of ref. 29 would not be the better way to communicate this.

In summary, I think a revision on the basis of these comments could be possible, I would suggest to extend the first part, possibly even to focus on the first part alone and to submit it to a more specialized journal.

List of main changes

1. We have added a new section "A unified phase diagram of gene expression and cellular growth" to discuss the extension of our model to more general situations, in which the limiting factors of gene expression are different.
2. We have shortened the discussion on the possible mechanisms of global extrinsic noise in the new version.
3. A new figure has been added (Figure 4 in the main text) which summarizes the proposed phase diagram of gene expression and cellular growth.

The main changes in the main text are shown in blue.

Reviewer #1 (Remarks to the Author):

The authors introduce a new coarse-grained model of stochastic gene expression, that seeks to explain the homeostasis of protein and mRNA concentrations in exponentially growing cells. This cannot be done with standard current gene expression models, which generally assume a fixed cell volume with constant transcription and translation rates that are proportional to the gene copy number. Based on an analysis of this model, the authors make a number of predictions about the mechanisms underlying the observed homeostasis in protein and mRNA concentrations. The results presented clearly make a contribution to the literature on modelling of stochastic gene expression.

We thank the reviewer for their careful reading and appreciating that our work contributes to the field.

Whether these results are sufficiently clearly explained or important to warrant publication in an interdisciplinary journal like Nature Communications I am unsure.

Major comments:

1. The authors do a rather poor job of explaining the contribution and importance of their work to readers outside the particular field of stochastic gene expression modeling. Many terms are used without being clearly defined (e.g. intrinsic versus extrinsic noise) and sentences such as “The homeostasis of protein and mRNA concentrations imply that there must be a regulatory mechanism in place to prevent the accumulation of noise over time and to maintain a bounded distribution of concentrations” are unnecessarily complicated and assume a fair degree of specialist knowledge on the part of the reader – what “noise” specifically could be being accumulated over time here, and what are the “bounded distributions” referred to (this is the first time such an expression is used in the paper)? I think the authors need to “zoom out” a little and begin the paper with a much more readable explanation of what they are trying to do.

Answer: We thank the reviewer for this useful suggestion. In the new version of the manuscript, we have elaborated on those points and attempted to make the terminology self-contained for readers outside the field of gene expression

Furthermore, following this comment by the reviewer as well as those of reviewer 2, we have made significant changes to the structure of the manuscript which we trust improves its readability and accessibility to readers outside the field:

1. We have improved the introduction significantly, and explained the motivation for our study in more detail.

2. As suggested by Reviewer 2, we removed most of the discussion on the possible mechanisms of global extrinsic noise from the manuscript since we trust this distracts

from the main takeaway of the work. Instead, we extended our main results to more general situations, as we briefly summarize now:

- In the first part of our model (Section: Model of stochastic gene expression), we consider the biological scenario in which both ribosomes and RNA polymerases (RNAPs) are limiting, i.e., neither DNA nor mRNA is saturated by the global machinery of the central dogma.

- In the last part of the new manuscript (Section: A unified phase diagram of gene expression and cellular growth), we consider the possibility that DNA can be saturated by RNAPs and that mRNA can be saturated by ribosomes. Assuming that each gene and mRNA has an upper limit of the number of RNAPs or ribosomes associated with them, we find that a transition from exponential growth of cell volume to linear growth emerges as the protein-to-DNA ratio increases. The linear growth regime occurs when both DNA and mRNA are saturated. This prediction can be tested (e.g., in bacterial or mammalian cells) by studying anomalously large cells (e.g., by inhibiting cell division).

Additionally, following the reviewer's comment, we added the following paragraph in the introduction to explain why there must be a regulatory mechanism for concentration homeostasis:

"Since the cell volume, protein copy number and mRNA copy number grow exponentially throughout the cell cycle, one may expect a sufficient condition to achieve a constant concentration is to let them grow with the same exponential growth rate. However, mathematical analysis suggests this is insufficient. Let us consider the logarithm of protein concentration c , which can be written as $\ln(c) = \ln(p) - \ln(V)$. Here p is the protein number and V is the cell volume. If one assumes the protein number and the cell volume grow exponentially but independently, with time-dependent exponential growth rates $\lambda_p(t)$ and $\lambda_V(t)$, respectively, the time derivative of the logarithm of concentration then obeys $d\ln(c)/dt \sim (\lambda_p(t) - \lambda_V(t))$. Even when the time-averaged growth rates of protein number and cell volume are equal, $\langle \lambda_p(t) \rangle = \langle \lambda_V(t) \rangle$, any fluctuations in the difference between them will accumulate and lead to a random walk behavior of the logarithm of concentration. The homeostasis of protein and mRNA concentrations implies that there must be a regulatory mechanism in place to prevent the accumulation of noise over time."

2. The authors develop a simple model and simulate / analyse it to make some predictions. As such, I am uncomfortable with their repeated use of the term "we show that". There is no data in the paper – the results are simply hypotheses (hopefully testable ones – this is not well discussed) that derive from the assumptions on which the model is based. If one agrees that the assumptions are reasonable, then the resulting predictions should be plausible, but they will only be "shown" to be true if/when they are confirmed experimentally (would the authors not agree?). It would be more accurate to use the term "our model predicts that...". Further the claim that "We verify our theoretical results using numerical simulations" is not accurate –

numerical simulations simply implement theoretical (mathematical) equations on a computer, they do not “verify” anything.

Answer: We thank the reviewer for this comment. In the new version, we have rephrased the phrasing accordingly.

In general, the paper needs a much more convincing discussion about how the various predictions made by the model could be tested and confirmed experimentally.

Answer: In the new version, we have elaborated on the relation between our model and experiments, and emphasized the model predictions, as we summarize below.

1. A central point regards the nature of cellular homeostasis. Our model predicts the homeostasis of mRNA and protein concentrations in an exponentially growing cell volume, which as discussed in the manuscript is a ubiquitous scenario in cellular growth. In contrast, the constant rate model would predict a constant mRNA number proportional to the gene copy number and linear growth of protein number, incompatible with various experimental observations. Our main innovation is to consider the limiting natures of RNA polymerase in transcription and ribosome in translation, which are also missing in the constant rate model. The limiting nature of ribosome leads to the exponential growth of cell volume and protein numbers, further inclusion of competition among RNA polymerases leads to the exponential growth of mRNA numbers.

2. Two additional predictions of our model are the proportionality between the mRNA number and the cell volume, and the independence of transcription rate of the genome copy number. These appear to be consistent with recent experiments on various organisms: we reproduce below experimental results on mammalian cells from Padovan-Merhar, *et al.*, *Molecular cell*, 2015 (Ref 19). The y axis is the mRNA number and the x axis is the cell volume. The slope represents the mRNA concentration. First, the experimental data shows a clear linear dependence of the mRNA number on the cell volume. Second, no significant changes of the slope before and after the genome replication (S phase) is observed, which implies a constant transcription rate independent of the genome copy number. Similar results are observed in other experiments (Kempe, *et al.*, *MBoC*, 2015 & Ietwsaart, *et al.*, *Cell systems*, 2017).

Following the reviewer's comment, we now discuss the above points regarding the shortcomings of the constant rate models in the manuscript (first paragraph in the Introduction and first paragraph in the section "Model of stochastic gene expression")

"Assuming a finite degradation rate of mRNAs and non-degradable proteins, these models lead to a constant mRNA number proportional to the gene copy number and linear growth of protein number (Marathe, *et al.*, 2012, Bierbaum & Klumpp, 2015, Jones, *et al.*, 2014), incompatible with the proportionality of mRNA and protein number to the exponentially growing cell volume"

"In constant rate models, the transcription rate per gene and the translation rate per mRNA are constant (Thattai & Van Oudenaarden, 2001, Paulsson, 2005, Shahrezaei & Swain, 2008) (Figure 1a). This implies that the gene (mRNA) number is the limiting factor in transcription (translation). Constant rate models predict a constant mRNA number proportional to the gene copy number and independent of the cell volume. However, experimental observations on plant and mammalian cells have revealed a proportionality between mRNA number and cell volume for cells with a constant genome copy number (Kempe *et al.*, 2015, Padovan-Merhar *et al.*, 2015, Ietswaart *et al.*, 2017}. Moreover, even comparing the cells before and after the genome replication (S phase), the proportionality coefficient between mRNA and cell volume does not exhibit any obvious change. In contrast, a constant transcription rate per gene would predict a doubled transcription rate after the replication of the whole genome, leading to a higher mRNA concentration. In one class of constant rate models (Marathe, *et al.*, 2012, Bierbaum & Klumpp, 2015, Cole & Luthey-Schulten, 2017}, a deterministic exponential growth of cell volume is explicitly considered. The resulting perturbation on the concentrations due to genome replication is suppressed in the long lifetime limit, but still significant for short lifetime molecules, e.g., mRNA (see Fig.1 in Ref. (Bierbaum & Klumpp, 2015))"

3. Another testable model prediction regards gene dosage effects arising from the finite duration of DNA replication. In the revised version we find their implications on the time-dependence of both transcription and translation throughout the cell cycle – which are directly testable experimentally. See the new discussion and proposed experimental test we added to Section "Effect of gene replication":

"The abrupt increase of ϕ_i corresponds to the replication of the specific gene i (Figure 3a) $\phi_i \rightarrow 2\phi_i$. However, as other genes are replicated, the relative fraction of gene i in the total genome decreases. This modulation propagates to the mRNA concentration which essentially tracks the dynamics of ϕ_i due to its short lifetime. The modulation of mRNA concentration affects the protein concentration as well, yet with a much smaller amplitude. These results can be tested experimentally by monitoring the DNA replication process and mRNA concentration simultaneously. We predict a quickly increasing mRNA concentration after the gene is replicated, followed by a gradual decrease of mRNA concentration until the next round of replication."

4. Importantly, as suggested by Reviewer 2, in the new version we explore scenarios where the assumptions regarding the limiting nature of RNA polymerase and ribosome are relaxed (as briefly mentioned in reply to Question 1). We predict a transition from exponential to linear growth of cell volume for sufficiently large cells where the limiting factors of transcription and translation may change, as we elaborate below.

We find that as the protein-to-DNA (PTD) ratio, γ , changes, three regimes of cellular growth are possible:

i) In Phase 1, neither DNA nor mRNA is saturated. RNA polymerases and ribosomes are respectively limiting for transcription and translation. In this phase, mRNA number, protein number and cell volume all grow exponentially. In the first part of our manuscript, we focus on this phase.

ii) In Phase 2, DNA is saturated while mRNA is not. The limiting factor in transcription becomes the gene number, while the limiting factor in translation is still ribosomes. A threshold value of protein-to-DNA ratio (γ_1) separates Phase 2 from Phase 1. In Phase 2, the production rate of mRNA becomes linear, but the production rate of protein is still exponential and the cell volume grows exponentially as well. The cell volume growth rate is the same as Phase 1, which is determined by the ribosomal fraction in the total proteome.

iii) In Phase 3, both DNA and mRNA become saturated. The limiting factors in transcription and translation are now the gene copy number and the mRNA copy number. A threshold value of protein-to-DNA ratio (γ_2) separates Phase 3 from Phase 2. In this phase, the cell volume grows linearly and the linear cell volume growth rate is proportional to the genome number, n_g . The three phases are summarized in the

following figure (shown as Figure 4 in the main text now). Expressions of the two thresholds are provided in the main text.

This extension of the model is discussed in a new section of the revised manuscript: "A unified phase diagram of gene expression and cellular growth".

This extension of our model provides concrete predictions on the growth dynamics of mRNA and protein number. One testable prediction regards the transition from exponential growth to linear growth of cell volume as the protein-to-DNA ratio increases. In fact, our colleagues from Harvard Medical School have performed related experiments (Xili Liu *et al.*, personal communications, 2018): they inhibit cell division and measure the growth of cell mass of mammalian cells. Indeed, they found a transition from exponential growth to linear growth of cell mass when the cell mass is larger than a threshold value, consistent with our theoretical prediction. In the new version, we have added a brief discussion of the experiment:

"We propose future experiments to study the potential transition from exponential to linear growth of cell volume, for example using filamentous *E. coli* by inhibiting cell

division and gene replication. Similar experiments can also be done for larger cells, e.g., mammalian cells, in which the transition from exponential growth to linear growth of cell volume may be easier to achieve. Preliminary results from experiments measuring the growth of cell mass of mammalian cells by inhibiting cell division indeed show a crossover from exponential growth to linear growth when the cell mass is above a threshold value (Xili Liu *et al.*, personal communications, 2018), consistent with our prediction."

Furthermore, in the new version, we also estimate the PTD ratio and the two values of the protein-to-DNA threshold ratios for *E. coli*:

"To gain some sense regarding the parameters associated with our proposed phase diagram, we estimate the PTD ratio of *E. coli*. Considering the typical cell volume of *E. coli* as $1 \mu\text{m}^3$, the protein density as 3×10^6 proteins/ μm^3 and the total number of protein-coding genes in *E. coli* as 4000 (Milo and Phillips, Cell biology by the numbers, 2015), we estimate the protein-to-DNA ratio for *E. coli* as $\gamma \sim 1000$. Estimates of the two threshold values of PTD ratios (see Methods) suggest that $\gamma_1 \sim 1000$ and $\gamma_2 \sim 20000$. We find that *E. coli* cells are typically in Phase 1, but not too far from Phase 2. "

3. The limitations of the proposed modeling approach need to be more clearly discussed. The authors consider only the case of constitutively expressed genes, where the only interactions between said genes are the competitions for ribosomes and RNAPs. This is a rather limited case, and it is not really clear whether the proposed approach can be extended to more realistic / complex scenarios. Other than for the purposes of investigating the specific questions considered in this paper, is there any particular advantage to the author's proposed modeling framework, over standard fixed-volume models (which certainly can take more complex interactions in to account)? If there is, it is not clear to me at the moment.

Answer: We trust that the above response partly answers the reviewer's question regarding the utility of our model. Nevertheless, the question regarding the extension of the model to the case where genes interact with each other is an excellent one. Within the previous version of our paper we indeed did not account for such scenarios explicitly, and essentially coarse-grained any additional layers of gene regulation into a modified value of the gene allocation fraction ϕ_i (controlling the expression level of this gene through affecting its relative standing in the competition for RNA polymerases with different genes). In fact, it is possible to extend our model to more realistic / complex situations where interactions between genes are present, e.g., through adding transcriptional regulation: $m_i \xrightarrow{k_0 \times \phi_i(c_j) \times n} m_i + 1$. Here the transcription rate of one type of mRNA (m_i) is affected by the protein concentration of another gene, c_j . To explore what may be lost in the aforementioned coarse-graining procedure, following the reviewer's comment we tested our model in a simplified scenario containing such regulation, as we now describe.

We considered an on-off transition of gene activation, a widely studied model in the

constant rate model:

Here, the gene copy number can transit between zero and a finite value with two constant rates. The gene activation and deactivation rate is set as $k_g^+ = k_{g0}c_{TF}$ and k_g^- . Here, c_{TF} is the concentration of transcription factor and k_{g0}, k_g^- are constant. The trajectories of cell volume, protein number, mRNA number and gene allocation fraction of one regulated gene are shown in the figure below (see details in Methods & Figure S1 in the SI). We remark that upon specific regulation, the protein copy number and mRNA copy number can show richer dynamics which may deviate from the exponential growth we find for constitutively expressed genes – see figure below. However, the cell volume growth (or total protein number) on a global level is not affected by those specific regulated genes since the sum of gene allocation fractions over the whole genome is precisely 1 (see top panel of figure below).

Following this comment, in the new version of manuscript, we have added a new section to the Methods on the discussion of the extension of our growing cell model to regulated gene expression (Methods: "Simulations of gene activation").

4. The authors should discuss the implications of the results in the following paper for their work, as it seems to be closely related:

<https://www.biorxiv.org/content/early/2018/02/18/267658>

Answer: We thank Reviewer 1 pointing out this important reference. We find the paper very interesting and indeed related to our work. The goal of the paper is to identify the sources of fluctuations in growth and how they propagate across the cellular machinery. This is quite different than our major goal, which is to understand the principles governing the homeostasis of mRNA and protein concentrations in exponentially growing cells. At the technical level, we summarize the main differences between their work and ours:

1. Although they introduce ribosomes as the limiting factor in translation, they do not consider competition between genes for RNA polymerases, i.e., they assume a constant transcription rate (potentially regulated). We find that their model is consistent with Phase 2 of our model, in which DNA becomes saturated but mRNA not. Importantly, their model cannot reproduce Phase I of our model, in which growth of cell volume/protein/mRNA is exponential. As mentioned earlier, data on various organisms (including mammalian, plant cells) appears to be consistent with this Phase (and not with Phase 2).
2. They modelled cell division based on replication initiation of DNA, which is related to section "Effect of gene replication" in our paper. Within Phase 2 of our model, the transcription rate is proportional to the gene copy number, which they seem not to take into account.

To summarize, their work and ours do not contradict each other yet are focusing on different aspects of cell growth and gene expression, and the details of the models differ. In the new version of our manuscript, we have cited this relevant reference and added a brief discussion of it.

Reviewer #2 (Remarks to the Author):

This paper addresses several questions related to gene expression experiments, in particular, the gene copy number and cell cycle dependence of mRNA and protein concentrations, the source of the extrinsic noise observed in the Taniguchi et al study (ref. 28), and the missing correlation between mRNA and protein seen in the same study. All these questions (possibly with the exception of the last one) are timely and of some importance, so a study of this type is very welcome. Nevertheless, I am not entirely convinced the importance of the paper is high enough to justify publication in a high-impact journal. The scope of what the authors try to address results is very wide, which has the unfortunate consequence that some issues that could have been nicely analyzed separately are not clearly distinguished and also previous work is not always recognized.

Overall, I find the first aspect, the gene copy number and cell cycle dependence the most intriguing of the 3 topics of the paper. The extrinsic noise part remains somewhat inconclusive, although plausible; and I do not see the point of the last part about correlations (basically, this is a remark that earlier work was flawed in a rather obvious way).

Here are a few more detailed comments:

1) Volume growth and cell cycle: The authors study two aspects of gene expression here that are often not included in simpler models: the growth of the cell volume and the competition of genes for machinery. They refer to previously models without the two as fixed-volume models. In my opinion, this is not correct, as some previous models have studied the first aspect (e.g. Jones et al, ref. 31, but also two papers by Bierbaum and Klumpp, *J Stat Phys* 2012 and *Phys Biol* 2015.) One result of these studies is that the protein concentration does not vary strongly over the cell cycle, but the mRNA concentration does, which is related to different half-lives. In the Jones et al. paper, the predicted variation seems to agree with experimental data, although they do not measure time series of the mRNA concentration directly, which would be needed for a definite answer. What is really new in this paper is that competition for machinery is also included, which in my understanding suppresses the small but systematic variations over the cell cycle that were seen in the earlier papers. For mRNA this would be a testable prediction where the models differ.

Answer: We first thank Reviewer 2 for pointing out two references that we did not cite in our previous version of manuscript (Marathe, *et al*, *J Stat Phys* 2012, Bierbaum & Klumpp, *Phys Biol* 2015). We read the two references carefully and realize that indeed they are very relevant for us. We agree with the reviewer that these two references consider an exponential cell volume growth explicitly and it is therefore inappropriate to call them fixed-volume models. In the new version of manuscript, we instead denote them as constant rate models because the transcription rate per gene and the translation rate per mRNA are constant. As the reviewer correctly points out, the key point distinguishes our works from these two papers is the introduction of the competition for the global machinery, which leads to exponential growth of mRNA

copy number, protein copy number and cell volume. Below we briefly summarize the main differences between our model and these previous models:

1. In these two papers, the authors assume a linear growth of protein number, and do not take into account the auto-catalytic nature of ribosomes. In our work, we consider the ribosome as the limiting factor in translation, therefore an exponential growth of protein number. As we discuss later (see the reply to Question 3), this may not always be true. In fact, as we explain in the newly added section to our manuscript, we find that an excess of ribosomes inside the cell (relative to the mRNA copy number) can lead to the saturation of mRNA and therefore a linear growth of protein.

2. In these two papers, the cell volume growth is assumed to be exponential and decoupled from the protein growth (in Marathe, *et al*, *J Stat Phys* 2012, a linear growth of cell volume is also considered). In contrast, in our model, we assume the cell volume is proportional to the sum of all proteins, and therefore tightly correlated with protein copy number, the regulation of which is a key focus of our work. This assumption is supported by experiments on various organisms. For example, in bacteria it has been shown that cellular dry mass density is very narrowly distributed (Kubitschek, *et al.*, *Journal of Bacteriology*, 1984; Basan, *et al.*, *Molecular Systems Biology*, 2015).

Another implication of this different assumption of the model regards the nature of the extrinsic noise: The extrinsic noise introduced by cell cycle effects in the aforementioned two papers is finite. In Phase 1 of our model (without considering density/ribosomal level fluctuations), the extrinsic noise is strictly zero, which can be seen by simulating an exponentially growing population. As shown in the figure below, the noise level is quantified by the squared coefficient of variation ($CV^2 = \text{variance}/\text{mean}^2$) of protein concentration, plotted as function of the mean protein number per cell volume (shown as Figure S3 in the SI).

If there is only intrinsic noise and no extrinsic noise, the noise level will keep decreasing as the mean increases, i.e., noise $\sim 1/\text{mean}$, which is exactly what we observe in the above figure. Note that as suggested by the reviewer, we removed most of the discussion concerning extrinsic noise, to focus on the main message of the

work.

3. We now discuss the connection of Jones, et al, *Science*, 2014 and our work. In their work, the authors adopt a commonly used model of stochastic gene expression: a constant transcription rate per gene copy number (they do not consider protein production in their work). The main cell cycle effect they consider as far as we can tell is a doubled transcription rate after the gene is replicated (in their paper, they consider mRNA copy number instead concentration, therefore cell volume growth is irrelevant). This contrasts with our assumption in the first part of our manuscript (Section "Model of stochastic gene expression" – which in the new version we refer to as “Phase 1”, see reply to question 3 below for extended discussion), in which we assume RNA polymerase is the limiting factor in transcription.

For certain organisms (including mammalian cells and plant cells), experimental observations support our assumption: for example, in the figure below, reproduced from Padovan-Merhar, *et al.*, *Molecular cell*, 2015 (Ref 19), the y axis is the mRNA number and the x axis is the cell volume. The constant rate models would predict a constant mRNA number proportional to the gene copy number. In contrast, our model in Phase 1 predicts a linear relation between mRNA number and cell volume even with a fixed gene number, consistent with the experimental observations: for those cells in the same cell-cycle phase, e.g., G1 phase, the linear relation between mRNA number and cell volume is still observed.

Furthermore, no significant changes of the slope before and after the genome replication (S phase) is observed, and similar results are observed in other experiments (Kempe, *et al.*, *MBoC*, 2015 & Ietwsaart, *et al.*, *Cell systems*, 2017). This observation is again consistent with Phase 1 of our model in the more general version we discuss in the new version.

In contrast, the constant rate models would predict a changing mRNA concentration due to genome replication. (see Marathe, *et al.*, *J Stat Phys* 2012).

Following the reviewer's comments in the new version of our model, we have extended the discussion on these previous models (first paragraph in the section "Model of stochastic gene expression"):

"In constant rate models, the transcription rate per gene and the translation rate per mRNA are constant (Thattai & Van Oudenaarden, 2001, Paulsson, 2005, Shahrezaei & Swain, 2008) (Figure 1a). This implies that the gene (mRNA) number is the limiting factor in transcription (translation). Constant rate models predict a constant mRNA number proportional to the gene copy number and independent of the cell volume. However, experimental observations on plant and mammalian cells have revealed a proportionality between mRNA number and cell volume for cells with a constant genome copy number (Kempe *et al.*, 2015, Padovan-Merhar *et al.*, 2015, Ietswaart *et al.*, 2017}. Moreover, even comparing the cells before and after the genome replication (S phase), the proportionality coefficient between mRNA and cell volume does not exhibit any obvious change. In contrast, a constant transcription rate per gene would predict a doubled transcription rate after the replication of the whole genome, leading to a higher mRNA concentration. In one class of constant rate models (Marathe, *et al.*, 2012, Bierbaum & Klumpp, 2015, Cole & Luthey-Schulten, 2017}, a deterministic exponential growth of cell volume is explicitly considered. The resulting perturbation on the concentrations due to genome replication is suppressed in the long lifetime limit, but still significant for short lifetime molecules, e.g., mRNA (see Fig.1 in Ref. (Bierbaum & Klumpp, 2015))"

Nonetheless, one may question the assumption regarding the limiting factors for transcription and translation. Experimental observations on fission yeast cells (Zhurinsky, *et al.*, *Current Biology*, 2010) are consistent with our assumption that RNA polymerase is limiting for transcription. However, the authors also found that in cell-cycle-arrested mutants exceeding a certain size, total transcription rates plateaued as DNA became limiting for transcription at low DNA-to- protein ratios. This result suggests that the gene number can be limiting as well in certain conditions.

In the new section "A unified phase diagram of gene expression and cellular growth", we extend our model to the possible phases that RNAP or ribosome may not be the limiting factor. We estimate the threshold value of protein-to-DNA ratio for *E. coli* to enter the phase that DNA starts to be saturated, and therefore the gene number becomes the limiting factor. We found that the estimated threshold value is not too far from the protein-to-DNA ratio of *E. coli*, so it is possible in some experiments, e.g., Jones, *et al.*, *Science*, 2014, the transcription rate per gene number is constant. See also our more detailed answer for Question 3.

Note that while experimental observations on *E. coli* (Scott, et al, *Nature*, 2010) and budding yeast (Metzl-Raz, et al, *eLife*, 2017) support our assumption that ribosomes are limiting for translation, in the new analysis we also consider the possibility that transcript number rather than ribosome number becomes limiting for translation.

2) I find the language used by the authors sometimes potentially misleading. They state that the number of ribosomes rather than the number of mRNA limits translation

(and correspondingly for mRNA). In my view, this is true on a global level but not for individual genes (and I think the authors agree with that, as this is what they formulate in their model). This distinction is very important and I think this should be phrased more clearly.

Answer: We agree with Reviewer 2 that for each individual gene, the translation rate is set both by the global ribosome number and the mRNA number of the specific gene (similar for transcription rate and RNA polymerase).

In fact, in the revised version we consider a scenario of a regulated gene, leading to a time-dependent transcription rate (also depending on the concentration of a transcription factor). The resulting dynamics of mRNA number and protein number are, as expected, different for regulated genes. However, the global behavior of cell volume and protein dynamics are unaffected, which are the focus of our work.

In the revised version of our manuscript, we have added a brief discussion to clarify this point:

" In fact, explicit gene regulation can also be included in our model (Methods), with a time-dependent g_i . In such scenarios, g_i may be a function of protein concentrations (for instance, the action of transcription factors modifies the transcription rate). Such models will lead to more complex dynamics of mRNA and protein concentrations. However, since we are interested in the global behavior of gene expression and cell volume growth, we do not focus on these complex regulations in this manuscript. Our conclusions regarding the exponential growth of mRNA and protein number for constitutively expressed genes and the exponential growth of cell volume on the global level are not affected by the dynamics of gene expression of particular genes."

3) While there is general agreement that ribosomes are limiting for translation in the sense that (almost) all ribosomes are active in translation and mRNAs compete for them, I am less convinced the same is true for RNA polymerases and genes. At least for *E. coli*, it has also been claimed that there is an excess of RNA polymerases (e.g. the model by Klumpp & Hwa 2008 and the experiments by Bakshi et al 2013). Given that protein concentrations become almost constant even with a gene copy number dependence of transcription, again mRNA should be the quantity to look at here. If a more general model lies somewhere between no competition as in the studies mentioned above and the perfect competition studied here, it would be interesting to see how big the effect of competition is.

Answer: We thank Reviewer 2 for this insightful comment. We agree that it may not be always true that RNA polymerase is the limiting factor in transcription. In the new version of our manuscript, we have extended our first part to include a new section "A unified phase diagram of gene expression and cellular growth". We propose that there are in general three phases of gene expression and cellular growth, and the transition between the different phases is controlled by the protein-to-DNA ratio:

i) In Phase 1, neither DNA nor mRNA is saturated. RNA polymerases and ribosomes

are respectively limiting for transcription and translation. In this phase, mRNA number, protein number and cell volume all grow exponentially. In the first part of our manuscript, we focus on this phase.

ii) In Phase 2, DNA is saturated while mRNA is not. The limiting factor in transcription becomes the gene number, while the limiting factor in translation is still ribosomes. A threshold value of protein-to-DNA ratio (γ_1) separates Phase 2 from Phase 1. In Phase 2, the production rate of mRNA becomes linear, but the production rate of protein is still exponential and the cell volume grows exponentially as well. The cell volume growth rate is the same as Phase 1, which is determined by the ribosomal fraction in the total proteome.

iii) In Phase 3, both DNA and mRNA become saturated. The limiting factors in transcription and translation are now the gene copy number and the mRNA copy number. A threshold value of protein-to-DNA ratio (γ_2) separates Phase 3 from Phase 2. In this phase, the cell volume grows linearly and the linear cell volume growth rate is proportional to the genome number, n_g . The three phases are summarized in the following figure (shown as Figure 4a in the main text now). Expressions of the two thresholds are provided in the main text.

Within this extension of our model, we provide concrete predictions on the growth dynamics of mRNA and protein number. One testable prediction regards the transition from exponential growth to linear growth of cell volume as the protein-to-DNA ratio increases.

In fact, our colleagues from Harvard Medical School have performed related experiments (Xili Liu *et al.*, personal communications, 2018): they inhibit cell division and measure the growth of cell mass of mammalian cells. Indeed, they found a transition from exponential growth to linear growth of cell mass when the cell mass is larger than a threshold value, consistent with our theoretical prediction. In the new version, we have added a brief discussion of the experiment:

"We propose future experiments to study the potential transition from exponential to linear growth of cell volume, for example using filamentous *E. coli* by inhibiting cell division and gene replication. Similar experiments can also be done for larger cells, e.g., mammalian cells, in which the transition from exponential growth to linear growth of cell volume may be easier to achieve. Preliminary results from experiments measuring the growth of cell mass of mammalian cells by inhibiting cell division indeed show a crossover from exponential growth to linear growth when the cell mass is above a threshold value (Xili Liu *et al.*, personal communications, 2018), consistent with our prediction."

Furthermore, in the new version, we also estimate the PTD ratio and the two values of the protein-to-DNA threshold ratios for *E. coli*:

"To gain some sense regarding the parameters associated with our proposed phase diagram, we estimate the PTD ratio of *E. coli*. Considering the typical cell volume of *E. coli* as $1 \mu\text{m}^3$, the protein density as 3×10^6 proteins/ μm^3 and the total number of protein-coding genes in *E. coli* as 4000 (Milo and Phillips, Cell biology by the numbers, 2015), we estimate the protein-to-DNA ratio for *E. coli* as $\gamma \sim 1000$. Estimates of the two threshold values of PTD ratios (see Methods) suggest that $\gamma_1 \sim 1000$ and $\gamma_2 \sim 20000$. We find that *E. coli* cells are typically in Phase 1, but not too far from Phase 2. "

4) Noise in ribosomal gene allocation: Here I tend to find the model plausible, but likely not a unique solution. Yes, this could contribute to the observed extrinsic noise, but couldn't other factors do that too? There is evidence for fluctuations in metabolism and distributions of growth rate. Why should ribosomes be the dominant factor here?

Answer: We agree with Reviewer 2 that there are certainly other possible factors that may contribute the extrinsic noise. Therefore, it is hard to conclude a unique solution to the extrinsic noise. In the new version of our manuscript, we have removed most of the discussions on the global extrinsic noise and focus more on the gene expression

models and its extension, following the suggestion of the reviewer.

5) mRNA-protein correlation: In my opinion, this discussion is quite artificial. It starts with the fact that in the Taniguchi et al paper (ref. 28), mRNA and protein copy numbers are compared. Their averages should be correlated if different conditions are considered, but the instantaneous numbers in individual cells should not be strongly correlated, since the two quantities have very different time scales. I think that the Taniguchi et al. paper (which other than that is an outstanding landmark study) has contributed to confusion as it suggests that this is a surprising observation, while at the same time it also demonstrates that not. Ref 29 discusses the correlations from ref. 28 further and basically shows that these low correlations are not consistent with a large class of stochastic models. The authors of the paper under review claim now that ref. 29 correlates numbers of mRNA (number per cell, i.e. variable volume) and protein concentration (number per constant volume) rather than numbers in constant volume as required by their relation. I have not checked that, but if this criticism of ref. 29 is correct and it looks like it is, then the analysis of ref. 29 is simply wrong and what the authors describe as a puzzle in the field does not exist. I am wondering whether a short comment to the journal of ref. 29 would not be the better way to communicate this.

Answer: We agree with Reviewer 2 that this point is not too profound and rather separated from the main point of our manuscript, which is the gene expression model incorporating RNA polymerase and ribosomes. For this reason, in the new version of our manuscript we only comment on this point briefly in the second to last paragraph of the section "Model of stochastic gene expression":

"Considering the cell cycle dependence of mRNA number and the homeostasis of protein concentration throughout the cell cycle, the experimental observation in *E. coli* showing negligible correlations between mRNA number and protein concentration (Taniguchi, *et al.*, 2010) seems to be a natural result of the cell cycle effect (Hilfinger, *et al.*, 2016)."

In summary, I think a revision on the basis of these comments could be possible, I would suggest to extend the first part, possibly even to focus on the first part alone and to submit it to a more specialized journal.

Answer: As noted, we have changed the structure as suggested Reviewer 2. We now focus on the gene expression model itself and removed most of the discussions on extrinsic noise. Furthermore, we have added a new section extending the model to other physiological states of cell growth.

Reviewers' comments:

Reviewer #2 (Remarks to the Author):

The authors have done an excellent job with the revision, and I commend them for "not taking no for an answer"! They have clearly thought hard about the reviewers' comments and made extensive changes to both the content and presentation of the paper. I think it's now much improved, more focussed, and the contribution is now clear. Happy to recommend acceptance.

Reviewer #3 (Remarks to the Author):

The paper has been revised significantly. The focus of the paper is now the theoretical discussion of the authors' gene expression model, which has been extended considerably, including new results on gene duplication (non-synchronously for different genes) and on situations where DNA or RNA content becomes limiting for gene expression. I think overall this improved the paper a lot.

Here are a few comments:

1) The authors distinguish the two scenarios in which either the machine (RNAP or ribosome) or the template (DNA or mRNA) is limiting for gene expression. My understanding is that in the case of ribosomes a limitation by the machine is well-established with the typical picture that almost all ribosomes are active in translation. In transcription things may be less clear, as there is excess RNAP stored in non-specific binding to DNA. Thus, the distinction may be less clear-cut as genes do not only compete among themselves but also with an external reservoir (which) is also doubled in genome replication. Due to the existence of such a reservoir and thus a fraction of non-transcribing RNAPs, the alternative to ribosome-like limitation is not simply saturation of genes, but more complicated.

2) Related to this consideration, my impression is that the phase diagram of fig. 4 may be too simple because it only considers one aspect of these systems (saturation).

3) The conclusion that DNA replication effects cannot explain the extrinsic noise is already seen with the model from refs. 26-28, where these effects are stronger as they persist over longer times. In ref. 28, this type of noise is therefore attributed to RNA polymerase availability fluctuations.

4) on p. 2 the authors compare their experimental observations with the constant rate per gene model, correctly noting that short lifetimes of the molecules are required for the observation of the systematic deviations from a constant concentration that is predicted by that model (but not by the authors' model). The authors should check whether the experimental observations they cite as support for a constant concentration actually fulfil this requirement.

Reviewer #3 (Remarks to the Author):

The paper has been revised significantly. The focus of the paper is now the theoretical discussion of the authors' gene expression model, which has been extended considerably, including new results on gene duplication (non-synchronously for different genes) and on situations where DNA or RNA content becomes limiting for gene expression. I think overall this improved the paper a lot.

We thank Reviewer 3's for their careful reading and appreciation of our revision.

Here are a few comments:

1) The authors distinguish the two scenarios in which either the machine (RNAP or ribosome) or the template (DNA or mRNA) is limiting for gene expression. My understanding is that in the case of ribosomes a limitation by the machine is well-established with the typical picture that almost all ribosomes are active in translation. In transcription things may be less clear, as there is excess RNAP stored in non-specific binding to DNA. Thus, the distinction may be less clear-cut as genes do not only compete among themselves but also with an external reservoir (which) is also doubled in genome replication. Due to the existence of such a reservoir and thus a fraction of non-transcribing RNAPs, the alternative to ribosome-like limitation is not simply saturation of genes, but more complicated.

Answer: We thank the reviewer for this important point. We agree with the reviewer that for bacteria the transcription kinetics is more complicated due to the nonspecific binding of RNA Polymerases (RNAPs) (note that for some eukaryotic cells, experiments suggest that nonspecific binding of RNAPs is suppressed, Killeen & Greenblatt, *Molecular and Cellular Biology*, 1992). Previous works (Klumpp & Hwa, *PNAS*, 2008; Bakshi, *et al.*, *Biophysical Journal*, 2013) have shown that in bacteria such as *E. coli* there are excess RNAPs nonspecifically bound to DNA and modeled their kinetics.

Following this comment, we consider a more detailed model in the Methods, taking account of the nonspecific binding of RNAPs. We consider the partitioning of RNAPs into four different classes for simplicity: 1) elongating RNAPs, n_e , 2) RNAPs bound to a promoter, n_p , 3) RNAPs nonspecifically bound to DNA, n_{ns} , 4) free RNAPs, n_{free} . We assume that the numbers of promoter-bound and nonspecifically bound RNAPs are determined by the concentration of free RNAPs in a Michaelis-Menten way (Klumpp & Hwa, *PNAS*, 2008, Patrick, *et al.*, *Biochimie*, 2015):

$$n_p = G \frac{c_{free}}{c_{free} + K_s},$$

$$n_{ns} = G g_{ns} \frac{c_{free}}{c_{free} + K_{ns}}.$$

Here c_{free} is the concentration of free RNAPs. K_s , K_{ns} are the Michaelis constants. G is the total number of genes and g_{ns} is the number of nonspecific binding sites per gene.

The number of elongating RNAPs is proportional to the number of RNAPs bound to promoter,

$$n_e = n_p k_{cat} \tau_0 = n_p \Lambda,$$

where k_{cat} is the transition rate from promoter-bound RNAPs to elongating RNAPs, τ_0 is the time for transcribing a gene, and $\Lambda = k_{cat} \tau_0$ is the ratio between the number of elongating RNAPs and promoter-bound RNAPs. The transcription rate for each gene is proportional to the number of elongating RNAPs on it, which in steady-state is also proportional to the number of promoter-bound RNAPs.

In a similar fashion to the aforementioned earlier works, we consider parameter regimes motivated by typical biological scenarios, in particular 1) the number of sites for nonspecific binding is much larger than the number of promoters, 2) nonspecific binding of RNAPs to DNA is much weaker than the specific binding of RNAPs to promoters, 3) the number of promoter-bound RNAPs is small compared with elongating RNAPs.

We derive a self-consistent equation for the fraction of free RNAPs in the total RNAP pool (Eq. (29) in the revised version).

First, we assume that the fraction of free RNAPs is small (Bakshi, *et al*, *Biophysical Journal*, 2013) and the concentration of free RNAPs is small compared with the specific binding Michaelis constant (later we discuss the results in the case where these assumptions are relaxed). We find that the fraction of elongating RNAPs (F_e) in the total RNAP pool is constant (i.e., it is independent of the cell cycle phase),

$$F_e = \frac{\Lambda}{1 + \Lambda + g_{ns} K_s / K_{ns}}.$$

To corroborate this result, we simulated a single lineage of growing cells using the full model (with partitioning of RNAPs and gene replication). We found that the fractions of elongating RNAPs and nonspecifically bound RNAPs during the cell cycle are approximately constant, with coefficient of variations of the order of 0.01, consistent with our theoretical analysis (see figure a below, shown as Figure S6a in the revised manuscript). The small modulations of the fraction of elongating and nonspecifically bound RNAPs during the cell cycle arise from the fact that free RNAPs still constitute a finite fraction of the total RNAP pool.

Within Phase 1 (we discuss the transition to Phase 2 below), the transcription rate (TR) for each gene is still proportional to the total number of RNAPs (n) and its gene allocation fraction (ϕ_i)

$$TR_i = \widetilde{k}_0 \times \phi_i \times n.$$

The competition between genes for RNAPs remains intact in Phase 1. In our original model, the transcription constant k_0 is the reciprocal of the time it takes to transcribe a gene. After we consider the nonspecifically bound RNAPs, it is renormalized by a constant factor $\widetilde{k}_0 = k_0 F_e$. Therefore, the introduction of nonspecifically bound RNAPs does not change our results qualitatively (but is important for quantitatively interpreting experiments).

Our numerical simulation indeed shows that the linear relation between mRNA number and cell volume is still valid to a good approximation, which is the key feature of Phase 1 (see figure b below, shown as Figure S6b in the revised manuscript).

In the new version of the manuscript this discussion now forms a new section in the Methods: "Effects of nonspecifically bound RNAPs", where the detailed calculations are performed.

In the main text of the new version, we have added a discussion at the end of the section "A unified phase diagram of gene expression and cellular growth" on the effects of nonspecifically bound RNAPs:

"It has been shown in bacteria that there are excess RNAPs nonspecifically bound to DNA (Klumpp & Hwa, 2008, Bakshi, *et al.*, 2013). In the Methods, we consider a modified model taking into account the partitioning of RNAPs to free RNAPs, elongating RNAPs, promoter-bound RNAPs and nonspecifically bound RNAPs. The transcription rate is determined by the concentration of free RNAPs through Michaelis-Menten kinetics (Klumpp & Hwa, 2008, Patrick, *et al.*, *Biochimie*, 2015).

We find that our conclusions remain intact with an approximately constant fraction of actively transcribing RNAPs in the total RNAPs for Phase 1 (Supplementary Fig. 6). The effect of nonspecifically bound RNAPs is therefore to renormalize the transcription constant k_0 in Phase 1 (Eq. (1a)) by a constant factor. "

2) Related to this consideration, my impression is that the phase diagram of fig. 4 may be too simple because it only considers one aspect of these systems (saturation).

Answer: We agree with the reviewer that our model is a coarse-grained model and we have neglected multiple details of gene expression. The motivation of our work is to connect the gap between various experimental observations on gene expression (linear scaling between mRNA number and cell volume, exponential growth of protein number and cell volume) and existing models (e.g., constant rate models). To this end, we attempted to consider a minimal model, which we constructed to be as simple as possible while in the meantime capturing the essence of the problem.

Following this comment and the previous comment, we have considered the effects of nonspecifically bound RNAPs on the phase transition from Phase 1 to Phase 2. Assuming the gene saturation is determined by the steric hindrance between elongating RNAPs, we find that the introduction of nonspecifically bound RNAPs does not change our previous result qualitatively and changes the threshold protein-to-DNA (PTD) ratio from Phase 1 to Phase 2, γ_1 , by a constant factor, $\tilde{\gamma}_1 = \gamma_1/F_e$, which is the inverse of the fraction of elongating RNAPs in the total RNAP pool.

We simulate the transition from Phase 1 to Phase 2 in the proposed experiment in which cell division is blocked, in the presence of nonspecific binding (see the figure below, shown as Figure S7 in the revised manuscript). At each point in time we solve the self-consistent equation of the partitioning of RNAPs. The plot shows both the deterministic dynamics of average mRNA number (black lines) as well as the results of the stochastic simulations (red/blue solid lines).

We compare the dynamics for two cells with different fixed genome sizes. Initially, the two cells are in Phase 1 and therefore have approximately equal mRNA number proportional to cell volume. When they exceed their respective threshold PTD ratio, the mRNA number begins to saturate and becomes limited by the gene number, therefore the cell with a twice larger genome size has twice more mRNAs (Phase 2). These results are consistent with previous experiments on fission yeast (Zhurinsky, *et al.*, *Current Biology*, 2010).

This discussion, including the associated calculations, is now contained in the section "Effect of nonspecifically bound RNAPs" in the Methods in new version.

We also studied the model behavior in the case where we relax the two assumptions made earlier, namely, that the fraction of free RNAPs is small and the concentration of free RNAPs is small compared with the specific binding Michaelis constant. We found that this introduces two alternative mechanisms of gene saturation.

One mechanism is the saturation of the fraction of free RNAPs. We denote F_{free} as the fraction of free RNAPs in the total RNAPs. The transcription rate of one gene can be saturated and proportional to its gene copy number when F_{free} becomes comparable to 1 from which we are able to derive a theoretical expression of the threshold protein-to-DNA ratio for F_{free} to saturate. Alternatively, gene saturation arises when the concentration of free RNAPs is comparable to the Michaelis constant, $c_{free} \approx K_s$, from which we also derive a theoretical expression of the threshold protein-to-DNA ratio.

The particular mechanism which drives the cell to Phase 2 (gene limiting) is the one with the smallest threshold.

The discussion related to the transition from Phase 1 to Phase 2 is now at the end of the section "Effects of nonspecifically bound RNAPs" in the Methods in new version.

In the main text of the new version, we have added a discussion at the end of the section "A unified phase diagram of gene expression and cellular growth" on the effects of nonspecifically bound RNAPs on the transition from Phase 1 to Phase 2:

"The transition from Phase 1 to Phase 2 is qualitatively unaffected (Supplementary Fig. 7) and the threshold PTD ratio γ_1 (Eq. (7)) from Phase 1 to Phase 2 is changed by a constant factor (Methods). We note that alternative mechanisms of gene saturation can occur upon introducing the different classes of RNAPs, through the saturation of free RNAPs and the Michaelis-Menten kinetics (Methods)."

3) The conclusion that DNA replication effects cannot explain the extrinsic noise is already seen with the model from refs. 26-28, where these effects are stronger as they persist over longer times. In ref. 28, this type of noise is therefore attributed to RNA polymerase availability fluctuations.

Answer: We agree with Reviewer 3 that the perturbation on protein concentration due to DNA replication is also quite small within constant rate models, as observed in Marathe, *et al*, *J Stat Phys* 2012, Bierbaum & Klumpp, *Phys Biol* 2015 and Jones, *et al.*, *Science*, 2014.

As shown in Figure 2B of Jones, *et al.*, *Science*, 2014, a systematic extrinsic noise is attributed to RNAP copy number fluctuations. In the new version of manuscript, we have added a comment on this point in the section "Effect of finite duration of gene replication":

"We note that a small extrinsic noise due to gene replication is also observed in constant rate models (Marathe, *et al.*, 2012, Bierbaum & Klumpp, 2015). Moreover, recent experiments and modeling have suggested a significant part of the extrinsic noise of mRNA expression level can be attributed to the fluctuations of RNAP copy number (Jones, *et al.*, 2014). Within our model, RNAP level fluctuations will lead to extrinsic noise in mRNA concentrations."

4) on p. 2 the authors compare their experimental observations with the constant rate per gene model, correctly noting that short lifetimes of the molecules are required for the observation of the systematic deviations from a constant concentration that is predicted by that model (but not by the authors' model). The authors should check whether the experimental observations they cite as support for a constant concentration actually fulfil this requirement.

Answer: We thank Reviewer 3 for pointing this important point. We have checked the three experimental papers we have cited on the linear scaling between mRNA number and cell volume. The only one that reported both mRNA life time and cell doubling time is the one by Ietswaart, *et al*, *Cell Systems*, 2017, in which the mRNA life time is about 6 hours and the cell doubling time is about 17 hours. This is comparable with the typical ratio between mRNA life time and cell doubling time for mammalian cells, which is about 0.3, larger than bacteria for which the mRNA life time-doubling time ratio can be smaller than 0.1 (Milo and Phillips, *Cell biology by the numbers*, 2015).

To test if such mRNA lifetimes are sufficiently short to allow us to distinguish between Phase 1 and 2 of our model (the latter equivalent to constant rate models with regards to transcription), we simulate our model for different mRNA lifetimes and compare the relation between mRNA number and cell volume. We consider two cases in which the mRNA lifetime-doubling time ratios are respectively 0.3 and 0.1. The doubling time is about 150 min, with the gene replication duration (C period for bacteria and S phase for eukaryotic cell) about 30 min starting near the middle of cell cycle. We find that given the typical mRNA lifetime-doubling time ratio of both

mammalian cells and bacteria, one can clearly distinguish Phase 1 and Phase 2 of our model. For Phase 1 (RNAP is limiting), a linear scaling between mRNA number and cell is observed, independent of the mRNA lifetime. For Phase 2, two plateaus for mRNA number corresponding to gene replication are observed. See the two figures below (Figure S4 in the new version of the manuscript). The dashed lines for Phase 1 are the theoretical prediction of our model (Eq. (5b)). The two dashed lines for Phase 2 correspond to the prediction based on a constant gene copy number before and after gene duplication.

In the new version of manuscript, we have added the following figures in the supplementary information and added a comment on this point in the section "A unified phase diagram of gene expression and cellular growth":

"In terms of transcription, our model in Phase 2 is equivalent to constant rate model and we have confirmed that for both bacteria and mammalian cells, the typical lifetime of mRNA is short enough compared with the doubling time to distinguish Phase 1 and Phase 2 (Supplementary Fig. 4)"

REVIEWERS' COMMENTS:

Reviewer #3 (Remarks to the Author):

The authors have done a convincing job answering to my comments and have added additional "controls"/additional analysis concerning the points in question. I recommend to accept this revised version of the paper.